# When Softmax Fails at the Top: Extreme-Value Corrections for InfoNCE

**Melihcan Erol** [* 1]   **Suat Evren** [* 1]   **Oktay Ozel** [2]   **Alexander Morgan** [1]   **Jongha Jon Ryu** [1]   **Lizhong Zheng** [1]

## Abstract

InfoNCE is the standard contrastive learning objective, but its softmax form is not only a computational convenience: it also encodes a statistical assumption about how the top-scoring example is selected. Using extreme value theory, we show that this assumption is often misaligned with the normalized embedding setting used in modern contrastive learning. Motivated by this mismatch, we propose WEINCE, a simple modification of InfoNCE that uses anchor-wise online batch statistics to blend the usual softmax logits with an endpoint shortfall correction, adding no trainable parameters. Across five vision benchmarks, WEINCE yields consistent improvements in frozen-feature evaluation. These results show that a more faithful statistical treatment of hard negatives can improve contrastive objectives.[1]

## 1. Introduction

Contrastive learning has become a central approach to self-supervised representation learning, and InfoNCE or closely related softmax-based objectives sit at the core of many influential methods in vision, language, and multimodal learning (Jaiswal et al., 2021; Le-Khac et al., 2020; van den Oord et al., 2018; Chen et al., 2020; He et al., 2020; Gao et al., 2022; Radford et al., 2021). The standard interpretation is that the model learns to assign high similarity to related views of the same example and low similarity to views from different examples (van den Oord et al., 2018; Chen et al., 2020; He et al., 2020). This approach has proved highly effective for learning transferable representations from unlabeled data (Jaiswal et al., 2021; Le-Khac et al., 2020; Chen et al., 2020; He et al., 2020; Gao et al., 2022; Radford et al., 2021).

In many of these systems, similarities are computed after normalizing the learned representations, typically through cosine similarity or an equivalent scaled inner product on the unit sphere (Chen et al., 2020; Wang & Isola, 2020; Gao et al., 2022; Radford et al., 2021). This places the loss in a bounded similarity regime. At the same time, prior work shows that contrastive learning is highly sensitive to the negative set. SimCLR reports strong gains from larger batch sizes and longer training (Chen et al., 2020), theoretical analyses emphasize the role of the number of negatives in representation quality (Saunshi et al., 2019), and several works show that hard negatives receive disproportionate weight under contrastive losses (Wang & Liu, 2021; Kalantidis et al., 2020; Robinson et al., 2021). Taken together, these observations suggest that the high-similarity tail of the negative distribution deserves special attention.

In most of this literature, the softmax in InfoNCE appears simply as part of the loss definition. Contrastive learning and InfoNCE are often justified through information-theoretic arguments, for example by showing that minimizing the loss maximizes a lower bound on mutual information (van den Oord et al., 2018; Chen et al., 2020; Lu et al., 2023; Tian et al., 2020; Poole et al., 2019; Alshammari et al., 2025; Ryu et al., 2026). In this paper, we show that the softmax implicitly assumes a specific assumption on the distribution of similarities. In particular, we borrow the insights from discrete choice theory, where softmax probabilities arise from the Plackett–Luce model for top-1 selection (Luce, 1959; Plackett, 1975). This provides a new perspective on InfoNCE: when the positive example wins against a set of negatives, the loss is implicitly fitting a statistical model for that winning event. This viewpoint lets us ask whether the hidden assumption behind softmax is compatible with the normalized embedding regime used in modern contrastive learning.

We answer this question negatively in the regime that matters most. The softmax assumption is not well aligned with normalized embeddings near the top of the similarity range, where the hardest negatives live. This leads to WEINCE (*Weibull-Enhanced InfoNCE*), a simple modification of InfoNCE that uses online batch statistics to interpolate between standard softmax logits and an endpoint shortfall correction. The interpolation is deliberate: it recovers InfoNCE when anchor-wise evidence for endpoint behavior is

---

[*]Equal contribution   [1]Massachusetts Insititute of Technology, Cambridge, Massachusetts   [2]Boston University, Boston, Massachusetts.   Correspondence to: Melihcan Erol <hsmerol@mit.edu>, Suat Evren <evrenis@mit.edu>.

*Proceedings of the 43rd International Conference on Machine Learning*, Seoul, South Korea. PMLR 306, 2026. Copyright 2026 by the author(s).

[1]Code: github.com/hsme98/weince.

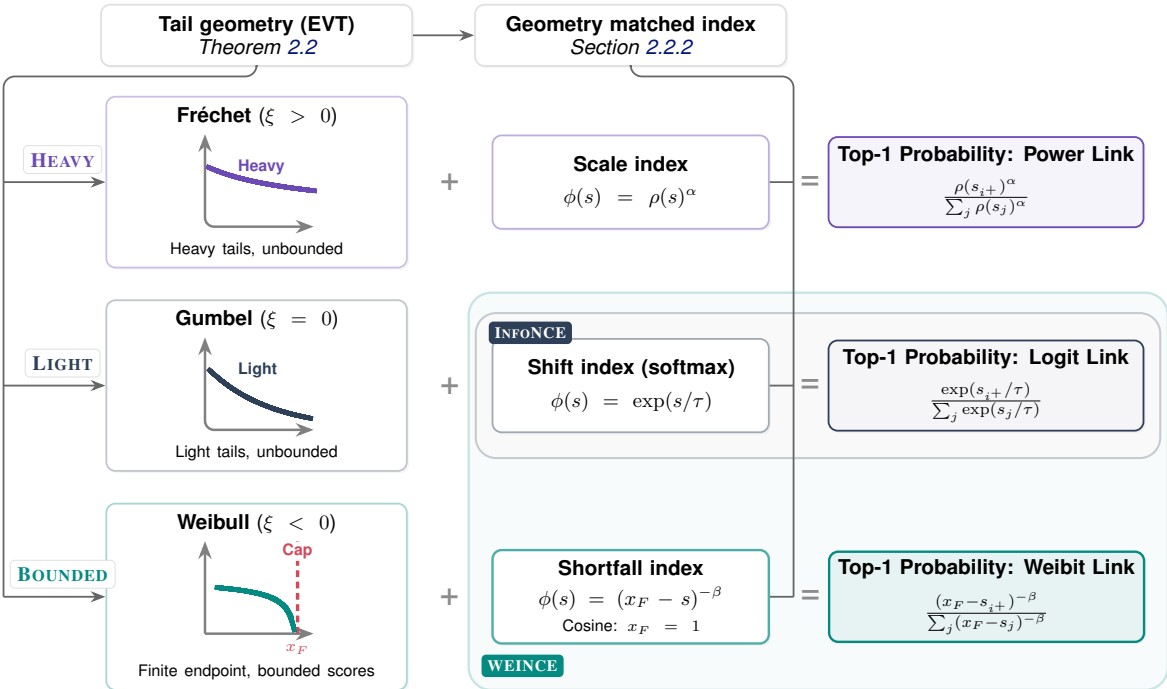

*Figure 1.* Extreme value theory (*Theorem 2.2*) yields three tail geometries. Pairing each geometry with a compatible index model gives a corresponding top-1 score. The Gumbel row recovers the softmax score used by InfoNCE. For bounded similarities, the Weibull row motivates a shortfall-based logit. WEINCE combines the softmax and shortfall logits through interpolation, so InfoNCE is recovered as the special case $\lambda = 0$.

weak and moves toward the Weibull shortfall link only when the current negative set is near the score cap. We support this view with theory, diagnostics, and experiments, and we show consistent improvements in frozen-feature evaluation across five vision benchmarks and SimCSE sentence embeddings.

Our contributions are threefold. First, we make explicit the statistical assumption hidden inside the standard InfoNCE softmax. Second, we show why this assumption becomes problematic in normalized embedding spaces and how the mismatch appears in likelihood and gradient-allocation diagnostics. Third, we introduce WEINCE, a practical replacement of InfoNCE that utilizes anchor-wise tail evidence to interpolate between softmax and Weibull shortfall logits.

## 2. Revisiting InfoNCE via Extreme-Value Theory

Self-supervised representation learning aims to learn an encoder $f$ from *unlabeled* data so that its representations transfer well to supervised downstream tasks, especially when labeled data is scarce. A standard pipeline first pretrains an encoder backbone with a self-supervised objective and then either fine-tunes the encoder or trains a lightweight prediction head using the available labels.

Contrastive learning implements self-supervision by con-

structing multiple views of the same underlying data point and treating these views as a positive pair, while treating views from other data points as negatives. A widely used contrastive objective in this setting is the InfoNCE loss.

**Definition 2.1** (InfoNCE loss). Let $(X, Y_0) \sim p_{XY}$ denote two correlated views (for example, an input and an augmentation), and let $Y_1, \ldots, Y_K \overset{\text{i.i.d.}}{\sim} p_X$ be negatives drawn independently of $(X, Y_0)$. Write $U = f(X)$ and $V_i = f(Y_i)$ for $i \in \{0, 1, \ldots, K\}$. The population InfoNCE loss is

$$\mathcal{L}(f) \triangleq -\mathbb{E}\left[\log \frac{\exp(s(U, V_0)/\tau)}{\sum_{i=0}^{K} \exp(s(U, V_i)/\tau)}\right], \quad (2.1)$$

with empirical counterpart

$$\widehat{\mathcal{L}}(f) \triangleq -\frac{1}{m} \sum_{j=1}^{m} \log \frac{\exp(s(u_j, v_{j0})/\tau)}{\sum_{i=0}^{K} \exp(s(u_j, v_{ji})/\tau)}. \quad (2.2)$$

where $s(\cdot, \cdot)$ is cosine similarity

$$s(u, v) = \frac{\langle u, v \rangle}{\|u\|_2 \|v\|_2},$$

and $\tau > 0$ is the temperature parameter.

## 2.1. InfoNCE as a Preference Model Estimation Objective

Each term in Equation (2.2) can be viewed as the negative log-likelihood of a *top-1 choice* event: for an anchor view $u_j$, the positive $v_{j0}$ is observed to "win" among the candidate set $\{v_{ji}\}_{i=0}^{K}$ (positive plus $K$ negatives). This connects InfoNCE to learning-to-rank and discrete choice models.[2]

**A general random-utility generative model.** Fix an anchor $j \in \{1, \ldots, m\}$ and let $\{s_{ji}\}_{i=0}^{K}$ denote the (latent) scores assigned to the $K+1$ candidates. A general way to model top-1 outcomes is through a *random utility model* in which each candidate $i$ is associated with a latent utility $U_{ji}$, drawn independently across candidates from a conditional family

$$U_{ji} \mid s_{ji} \sim P_{U \mid S = s_{ji}}, \qquad i \in \{0, 1, \ldots, K\},$$

and the observed outcome reveals only the winner

$$I_j = \arg\max_{0 \leq i \leq K} U_{ji}.$$

Under this lens, a contrastive dataset provides outcomes $\mathcal{I} \triangleq \{I_j\}_{j=1}^{m}$, and the learning problem is to parameterize the scores via an encoder and maximize the likelihood of the observed winners. Since the candidate indices are arbitrary up to relabeling, we may, without loss of generality, index the positive candidate as 0 for every anchor.

**Recovering InfoNCE as a special case.** InfoNCE corresponds to a particular *parametric* choice of the conditional model $P_{U \mid S}$, namely an *additive* random-utility model with i.i.d. Gumbel noise:

$$U_{ji} = s_{ji} + \epsilon_{ji}, \qquad \epsilon_{ji} \overset{\text{i.i.d.}}{\sim} \text{Gumbel}(0, \tau). \quad (2.3)$$

Under Equation (2.3), the top-1 probability takes the softmax form

$$\mathbb{P}\big(I_j = k \mid \{s_{ji}\}_{i=0}^{K}\big) = \frac{\exp(s_{jk}/\tau)}{\sum_{i=0}^{K} \exp(s_{ji}/\tau)}.$$

We parameterize scores using an encoder $f : \mathcal{V} \to \mathbb{R}^d$ via cosine similarity,

$$s_{ji} = \frac{\langle f(u_j), f(v_{ji}) \rangle}{\|f(u_j)\|_2 \|f(v_{ji})\|_2},$$

so the log-likelihood of observing $\mathcal{I}$ becomes

$$\begin{aligned}
\ell(f) &\triangleq \log \mathbb{P}(\mathcal{I}) \\
&= \sum_{j=1}^{m} \log \mathbb{P}(I_j = 0) \\
&= \sum_{j=1}^{m} \log \frac{\exp(s(u_j, v_{j0})/\tau)}{\sum_{i=0}^{K} \exp(s(u_j, v_{ji})/\tau)}. \quad (2.4)
\end{aligned}$$

---

[2]In the classical parametric case this reduces to the Plackett–Luce (multinomial logit) model.

Up to the factor $1/m$, the negative log-likelihood $-\ell(f)$ is exactly the empirical InfoNCE objective in Equation (2.2).

**Implicit assumptions behind the softmax link.** This likelihood view highlights what InfoNCE implicitly assumes about how top-1 outcomes are generated. The softmax form is not only tied to the *Gumbel* distributional choice, but also to the *additive-in-score* structure in Equation (2.3). More generally, however, the latent "utility" driving which candidate wins need not be an additive perturbation of the score: one could instead posit an unknown conditional model $P_{U \mid S}$ (potentially non-additive, heteroskedastic, or operating in a different tail coordinate altogether). This leads to the guiding question of the paper:

> *When is the softmax top-1 likelihood implied by Equation* (2.3) *a good approximation to the top-1 outcomes generated by an unknown conditional utility model $P_{U \mid S}$, and what replaces it when it is not?*

At this level of generality, the question is underdetermined: there are many ways to specify $P_{U \mid S}$, and different choices induce different top-1 link functions. To obtain principled guidance, we turn to asymptotics.[3] In modern contrastive learning, the number of negatives $K$ can be large, and the top-1 event is governed by the extreme behavior of $\max_i U_{ji}$, or equivalently by the tail behavior of the utility family. Just as the CLT provides a universal limit theory for sums, extreme value theory (EVT) (De Haan & Ferreira, 2006) provides a universal limit theory for maxima.

This motivates our next step: analyze the asymptotic regime where $K \to \infty$, using EVT to identify which aspects of $P_{U \mid S}$ matter for top-1 probabilities and to derive geometry-consistent alternatives to the softmax link.

## 2.2. Understanding Large-$K$ Behavior of InfoNCE via Extreme Value Theory

### 2.2.1. A TAIL GEOMETRY TRICHOTOMY

We now turn to extreme value theory (EVT), which characterizes the distributional behavior of maxima. The foundational result is the Fisher–Tippett–Gnedenko theorem (Gnedenko, 1943), which can be understood as a central limit theorem for maxima.

**Theorem 2.2** (Fisher–Tippett–Gnedenko). *Let $X_1, X_2, \ldots$ be i.i.d. random variables with common CDF $F$, and let*

---

[3]A recurring theme in statistics is that large-sample or high-dimensional limits reveal *universal* structure. Classical inference for sums rests on the central limit theorem (e.g., (van der Vaart, 1998)), while the Tracy–Widom law is a canonical universality result for the fluctuations of extreme eigenvalues of large random matrices (e.g., (Tracy & Widom, 1994; Anderson et al., 2010; Johnstone, 2001)).

$M_n = \max_{1 \le i \le n} X_i$. *If there exist normalizing sequences $a_n > 0$ and $b_n \in \mathbb{R}$ such that*

$$\mathbb{P}\left(\frac{M_n - b_n}{a_n} \le x\right) \to G(x)$$

*for some non-degenerate CDF $G$, then on any compact set $K \subset \{x : G(x) \in (0,1)\}$ there exists $\xi \in \mathbb{R}$ such that*

$$\frac{\bar{F}(b_n + a_n x)}{\bar{F}(b_n)} \to \nu_\xi(x), \tag{2.5}$$

*where $\bar{F} = 1 - F$ and*

$$\nu_\xi(x) = \begin{cases} \exp(-x), & x \in \mathbb{R}, & \xi = 0, \\ (1 + \xi x)^{-1/\xi}, & 1 + \xi x > 0, & \xi \ne 0. \end{cases}$$

This single parameter $\xi$ describes three qualitatively different tail geometries (see Figure 2):

1. **Fréchet ($\xi > 0$):** heavy tails (power law type) with unbounded right tail.

2. **Gumbel ($\xi = 0$):** rapidly or exponentially decaying tails under the appropriate normalization; the limiting law has support on all $\mathbb{R}$.

3. **Weibull ($\xi < 0$):** finite right endpoint with regular variation in the shortfall to that endpoint; after affine normalization the support is bounded above.

Strictly speaking, a domain of attraction is a property of a tail law, not of one realized finite set of scores. Moreover, bounded support alone does not force the Weibull domain: finite-endpoint distributions with sufficiently rapid decay near the endpoint can still be Gumbel-type. Thus, for bounded cosine similarities, the relevant question is local and anchor-dependent: whether the active hard-negative tail behaves more like a translation/Gumbel tail or like an endpoint shortfall/Weibull tail.

*Remark* 2.3. An equivalent statement of the theorem is that the limit law $G$ belongs to the generalized extreme value family. The theorem does not identify the absolute tail scale of $F$ since the normalization sequences $(a_n, b_n)$ absorb that. What remains universal is the relative tail geometry captured by the tail ratio limit (2.5).

**An extension to non-identical utilities.** For our purposes, the candidates need not be identically distributed, because their utility laws depend on their scores. We therefore use a non-identically distributed extension of the FTG tail-ratio characterization.

The formal statement and proof are given in Appendix A. The result requires a mild homogeneity condition that rules out pathological index orderings; this is natural in our setting because negative examples are randomly drawn without replacement.

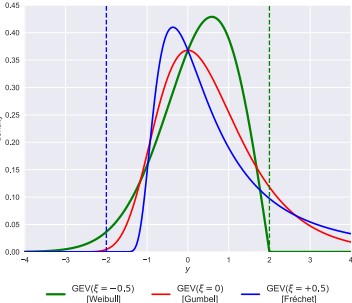

*Figure 2.* Examples of distributions exhibiting the three tail geometries (Fréchet, Gumbel, Weibull) that arise as limits for normalized maxima under Theorem 2.2.

### 2.2.2. CANONICAL INDEX MODELS GENERATING A TAIL GEOMETRY

In the previous section, we show that, in the large candidate regime, maxima exhibit one of three universal tail geometries (Fréchet, Gumbel, or Weibull). This suggests a modeling principle that we use throughout the paper: rather than committing a priori to a specific parametric form such as additive Gumbel noise as in Equation (2.3), we can first ask which tail geometry the latent utilities plausibly follow and then use a ranking model that is compatible with that geometry. Equivalently, each admissible tail geometry can be paired with a canonical random utility model whose induced maxima lie in that geometry. Since the observed label is a top-1 outcome, selecting a model with mismatched effective geometry can distort the relative probabilities in the extreme tail and can be statistically inefficient for estimating the underlying scores from winner observations.[4]

These canonical ranking models can be viewed as generalizations of the Plackett–Luce model in Equation (2.3), obtained by working in an appropriate tail coordinate. Concretely, there exists a monotone transform $T$ and an index map $\eta(\cdot)$ such that utilities satisfy

$$T(X_i) = \eta(s_i) + \varepsilon_i, \tag{2.6}$$

where $\varepsilon_i$ is candidate independent noise. Inverting the tail coordinate yields three representative forms:

(a) **Multiplicative utility (Fréchet):** $X_i = \rho(s_i)\, Z_i$.

(b) **Additive utility (Gumbel):** $X_i = s_i + \epsilon_i$.

(c) **Shortfall to endpoint (Weibull):** $X_i = b - q(s_i)\, W_i$.

Here $Z_i, \epsilon_i, W_i$ are i.i.d. noise variables whose tails lie in the Fréchet, Gumbel, and Weibull domains, respectively.

---

[4]This is the standard quasi maximum likelihood interpretation formalized in Theorem C.1; see also (White, 1982; Gourieroux et al., 1984; van der Vaart, 1998).

*Table 1.* **Tail geometry ($\xi$) and extreme value invariance.** A geometry matched indexing becomes additive in the tail coordinate: $T(X_s) = \eta(s) + \varepsilon$ with candidate independent noise $\varepsilon$ (for example, $\varepsilon = \epsilon$ in the Gumbel row, $\varepsilon = \log Z$ in the Fréchet row, and $\varepsilon = -\log W$ in the Weibull row). Here $\alpha = 1/\xi > 0$ in the Fréchet case and $\beta = -1/\xi > 0$ in the Weibull case.

| Geometry | Natural action | Tail coordinate $T(x)$ | Index statistic $\eta(s)$ | Canonical model | Top 1 rule |
|---|---|---|---|---|---|
| $\xi = 0$ (Gumbel) | translation | $T(x) = x$ | $\eta(s) = s$ | $X_s = s + \epsilon$ | $p_i \propto \exp\big(s_i/a_n\big)$ |
| $\xi > 0$ (Fréchet) | scaling | $T(x) = \log x$ | $\eta(s) = \log \rho(s)$ | $X_s = \rho(s)\,Z$ | $p_i \propto \rho(s_i)^{\alpha}$ |
| $\xi < 0$ (Weibull) | endpoint shortfall | $T(x) = -\log(b - x)$ | $\eta(s) = -\log q(s)$ | $b - X_s = q(s)\,W$ | $p_i \propto q(s_i)^{-\beta}$ |

### 2.2.3. ASYMPTOTICALLY CORRECT LINK FUNCTIONS

Recall the general formulation $U \mid S = s \sim P_{U|S=s}$ introduced earlier. Let $F_s$ be the CDF of $U \mid S = s$ and $\bar{F}_s(t) \triangleq \mathbb{P}(U > t \mid S = s)$ its survival function. The next result shows that, for large candidate sets, the top-1 selection probabilities depend on $P_{U|S}$ only through these tail probabilities evaluated at the normalization level $b_n$ from Theorem A.2. The assumptions and proof are in Appendix B.

**Theorem 2.4** (Tail mass proportionality). *Let $b_n \in \mathbb{R}$ and $a_n > 0$ be as in Theorem A.2. Then under Assumption B.1, uniformly in $1 \leq i \leq n$,*

$$p_i \triangleq \mathbb{P}(I_n = i \mid \boldsymbol{s}_n) = \frac{\bar{F}_{s_i}(b_n)}{\sum_{r=1}^{n} \bar{F}_{s_r}(b_n)}\,(1 + o(1)).$$

Theorem 2.4 reduces the large $K$ top-1 link induced by $P_{U|S}$ to the tail weights $\bar{F}_{s_i}(b_n) = \mathbb{P}(U > b_n \mid S = s_i)$. The FTG tail ratio characterization (Theorem 2.2) implies that, within a fixed EVT domain of attraction (equivalently, for a fixed tail geometry parameter $\xi$), these tail weights have universal asymptotic ratios. In Theorem B.2 we make this dependence on $\xi$ explicit, yielding geometry-dependent top-1 links. Evaluating the resulting tail weights for the canonical models in Section 2.2.2 gives the closed form probabilities in Table 1. The table should be read conditionally: different anchors, training stages, or negative pools may exhibit different effective tail geometries, and a finite minibatch cannot identify the asymptotic domain with certainty. Thus the theory motivates an adaptive link, rather than a global replacement of softmax by a single alternative. In the next section, we test whether winner outcomes observed during InfoNCE training show systematic endpoint evidence beyond the softmax link implied by Equation (2.3).

### 2.3. InfoNCE May Be Misspecified for Contrastive Representation Learning

Recall from Equation (2.3) that InfoNCE coincides with maximum likelihood when utilities follow an additive random utility model with i.i.d. Gumbel noise, which yields the softmax top-1 link. The discussion in Section 2.2 suggests that this is only one possible effective regime: even when the conditional utility model $P_{U|S}$ is unknown, the winner

event for large candidate sets is governed by tail behavior, and under approximate max-stability it must fall into one of three tail geometries indexed by $\xi$ (Gumbel $\xi = 0$, Fréchet $\xi > 0$, Weibull $\xi < 0$). From this asymptotic standpoint, diagnosing misspecification means asking how much endpoint geometry is active, and in which choice sets, rather than assigning one geometry globally. This leads to the following empirical question:

> *Is there systematic endpoint evidence beyond the softmax/Gumbel link, and when does it become important for top-1 prediction?*

#### 2.3.1. EMPIRICAL EVIDENCE: ENDPOINT BEHAVIOR IN THE SCORE TAIL

We begin with a direct look at a representative aggregate tail of high negative similarities. Applying the standard Peaks-Over-Threshold (POT) methodology to cosine similarities from a frozen encoder (Figure 3), we fit a Generalized Pareto distribution to exceedances above a high threshold. For this aggregate diagnostic, the fitted shape parameter is $\hat{\xi} = -0.39$, indicating Weibull-type endpoint behavior ($\xi < 0$) in the near-cap tail. This establishes an endpoint component in the hard-negative regime, but it does not imply that every finite anchor choice set should be modeled by a pure Weibull link. The Gumbel special case ($\xi = 0$) is a poor upper-tail fit: its density (red dashed in Section 2.3.1) assigns positive mass beyond the score ceiling, while the Weibull-GPD (black dashed) correctly drops to zero at the cap. The QQ plot (Section 2.3.1) gives the same message: the Weibull-GPD quantiles lie close to the diagonal, while the Gumbel deviates sharply in the upper tail.

#### 2.3.2. A LIKELIHOOD-BASED LINK SELECTION TEST

The POT analysis is marginal and tail-level; it does not decide whether every anchor should use a pure endpoint link. We now ask how much this component matters for top-1 prediction relative to the standard translation/Gumbel coordinate. We compare the standard softmax link (translation coordinate) against a nested one-parameter family that interpolates between the translation coordinate and an endpoint (shortfall) coordinate. Here, since heavy tail (Fréchet) behavior is not a natural hypothesis here because cosine similarity scores are bounded above ($s \leq 1$), we focus on

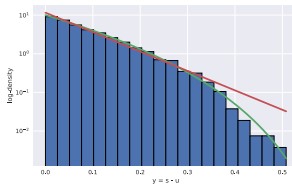 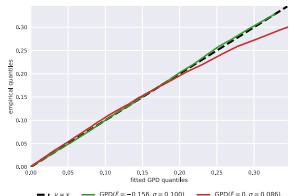

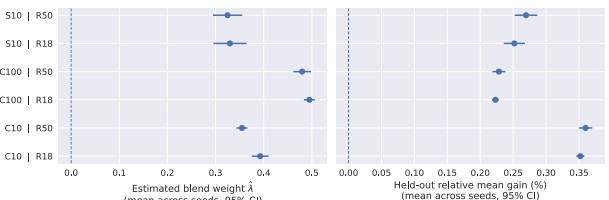

*Figure 3.* POT analysis of a representative aggregate score tail from a frozen encoder ($\hat{\xi} = -0.39$). **Left:** Exceedance density. Weibull-GPD (black) drops to zero at the cap (green); Gumbel-GPD (red) does not. **Right:** QQ plot. Weibull-GPD (blue) matches the upper tail; Gumbel (red) overestimates upper quantiles.

*Figure 4.* Likelihood-based link selection across datasets and backbones (R18: ResNet18; R50: ResNet50; C10/100: CI-FAR10/CIFAR100; S10: STL10). Left: estimated blend weight $\hat{\lambda}$ (mean across seeds, 95% confidence interval). Right: held-out mean log likelihood gain $\overline{\Delta}$ (nats per anchor; mean across seeds, 95% confidence interval). Numeric values are reported in Table 4 in Appendix D; epoch-wise trends (Figure 9) are provided in Appendix D.

translation (Gumbel) versus endpoint (Weibull) regimes in the main experiments. We defer an ablation of the Fréchet case to Appendix D.6.

Concretely, for each frozen checkpoint, we build an evaluation bank consisting of 32 batches of size $B = 256$, yielding $2B$ augmented views per batch. For each anchor view $i$ in a batch, the candidate set $\mathcal{N}_i$ is the other $2B - 1$ views in the same batch, and the observed winner is its paired positive $i^+$. We compute cosine similarity scores $s_{ik}$ for $k \in \mathcal{N}_i$ and treat the labels $\{i^+\}$ as top-1 outcomes. This evaluation does not require access to latent utilities.

We split the evaluation bank into held-in batches (used to fit link parameters) and held-out batches (used only for reporting). We compare:

- $\mathcal{H}_0$ **(softmax / Gumbel link):** Under the additive random utility model with i.i.d. Gumbel noise, the top-1 probability is the softmax link

$$p_0(Y_i = j \mid \{s_{ik}\}_{k \in \mathcal{N}_i}) = \frac{\exp(s_{ij}/\tau_0)}{\sum_{k \in \mathcal{N}_i} \exp(s_{ik}/\tau_0)}. \quad (2.7)$$

  We fit the temperature $\tau_0 > 0$ by maximizing the held-in top-1 log likelihood.

- $\mathcal{H}_1$ **(blended tail coordinate link):** To capture endpoint geometry, define the shortfall coordinate $g(x) \triangleq -\log(1 - x)$ for $x < 1$ (cosine endpoint $b = 1$), and the nested transform

$$T_\lambda(x) \triangleq (1 - \lambda)x + \lambda g(x), \qquad \lambda \in [0, 1]. \quad (2.8)$$

  The induced top-1 link is again softmax,

$$p_1(Y_i = j \mid \{s_{ik}\}_{k \in \mathcal{N}_i}) = \frac{\exp(T_\lambda(s_{ij})/\tau_1)}{\sum_{k \in \mathcal{N}_i} \exp(T_\lambda(s_{ik})/\tau_1)}. \quad (2.9)$$

  This family is nested since $\lambda = 0$ recovers $\mathcal{H}_0$, while $\lambda = 1$ gives the pure shortfall coordinate. Intermediate values quantify a partial endpoint component rather than an all-or-nothing replacement of softmax. We fit $(\lambda, \tau_1)$ on held-in batches by maximizing the top-1 log likelihood.

**Evaluation metric.** On held-out batches we report the mean log likelihood gain per anchor,

$$\overline{\Delta} = \mathbb{E}\big[\log p_1(Y_i = i^+) - \log p_0(Y_i = i^+)\big],$$

estimated by the held-out sample average (see Appendix D for details). Since $\overline{\Delta}$ is a mean log ratio, $\exp(\overline{\Delta})$ is the ratio of geometric mean probabilities assigned to the positive under $\mathcal{H}_1$ versus $\mathcal{H}_0$. We assess significance using a one-sided clustered $t$ test over batches, and we also report bootstrap-based confidence intervals (Appendix D).

**Results.** Across CIFAR-10, CIFAR-100, and STL-10, and across ResNet-18 and ResNet-50 backbones, the fitted blend weight is consistently nonzero but also far from one ($\hat{\lambda} \approx 0.33$ to $0.50$), indicating a systematic endpoint-shortfall component beyond pure translation geometry, rather than a wholesale replacement of softmax (Figure 4). This is consistent with the FTG view: many anchors contribute near-cap hard negatives, while others remain translation-like or statistically ambiguous. The held-out improvement is substantial ($\overline{\Delta} \approx 0.22$ to $0.36$ nats per anchor), corresponding to a geometric mean increase in assigned probability of $\exp(\overline{\Delta}) \approx 1.25$ to $1.43$. The gain is statistically significant under batch-clustered testing and remains stable across training epochs (Appendix D).

Additional ablations in Appendix D decouple choice-set hardness from negative-pool size. Including easier negatives dilutes both $\hat{\lambda}$ and $\overline{\Delta}$, while increasing the pool from which a fixed hard set is selected drives $\hat{\lambda}$ upward. This matches the EVT prediction that bounded-endpoint effects become stronger as harder near-cap competitors become available.

**Implication for training.** The link-selection test is more than a diagnostic of held-out likelihood. A consistently

nonzero endpoint weight means that a substantial fraction of anchors exhibit Weibull endpoint behavior, so a pure softmax link gives a statistically misspecified winner model on those anchors. In the quasi-MLE sense, softmax then targets the closest member of the wrong link family rather than the true top-1 law. This misspecification can lead to unfavorable gradient allocation, because contrastive training places gradient mass according to the fitted winner probabilities. We give a formal misspecification statement, a synthetic noise-floor illustration, and the risk-versus-gradient diagnostic in Appendix C. The method below uses this evidence constructively: rather than replacing softmax wholesale, WEINCE we propose below adapts the correction anchor by anchor.

## 3. WEINCE: Correcting Softmax at the Extremes

Section 2.3 shows that bounded-similarity top-1 outcomes often contain an endpoint shortfall component, especially for near-cap hard negatives. Because finite minibatches mix endpoint-like, translation-like, and ambiguous anchors, we use an adaptive objective that corrects InfoNCE logits only when anchor-wise tail evidence supports it. The full training step is summarized in Algorithm 1; implementation details are deferred to Appendix E.

**Logit level tail interpolation.** Consider a batch of $2N$ $\ell_2$ normalized embeddings $\{z_i\}_{i=1}^{2N}$ and cosine scores $s_{ij} \triangleq z_i^\top z_j \in [-1, 1]$. For each anchor $i$, let $i^+$ denote its paired positive and let $\mathcal{N}_i = \{1, \ldots, 2N\} \setminus \{i\}$ be the candidate set (positive plus negatives).

InfoNCE uses Plackett–Luce (softmax) logits

$$\ell_{ij}^{\mathrm{PL}} \triangleq \frac{s_{ij}}{\tau},$$

while a Weibull shortfall model suggests logits of the form

$$\ell_{ij}^{\mathrm{W}} \triangleq -\beta \log(x_F - s_{ij}) \qquad (s_{ij} < 1).$$

with $x_F = 1$ for cosine similarity, according to the tail-index model given in Equation (2.6) and Table 1. We define interpolated logits

$$\ell_{ij} \triangleq (1 - \lambda_i)\, \ell_{ij}^{\mathrm{PL}} + \lambda_i\, \ell_{ij}^{\mathrm{W}}, \qquad \lambda_i \in [0, 1], \quad (3.1)$$

where $\lambda_i$ is an anchor-wise interpolation weight and $\beta$ is estimated online (anchor-wise in our implementation, as described below). This nesting is intentional: $\lambda_i = 0$ recovers vanilla InfoNCE, $\lambda_i = 1$ gives a pure endpoint-shortfall link, and intermediate values represent anchors for which the tail evidence is mixed. The resulting loss is standard cross entropy:

$$\mathcal{L}_{\mathrm{WEINCE}}(\theta) \triangleq -\frac{1}{2N} \sum_{i=1}^{2N} \log \frac{\exp(\ell_{i,i^+})}{\sum_{j \in \mathcal{N}_i} \exp(\ell_{ij})}. \quad (3.2)$$

In practice, $\lambda_i$ is estimated separately for each anchor rather than fixed globally, so ambiguous anchors remain close to InfoNCE while endpoint-like anchors receive larger shortfall corrections.

**Endpoint shortfall for cosine scores.** For bounded cosine similarity the endpoint is $x_F = 1$. The Weibull row of Table 1 gives $p_j \propto q(s_{ij})^{-\beta}$. Locally near the endpoint, any differentiable shortfall scale with $q(s) \to 0$ is equivalent up to a multiplicative constant, so we use $q(s) = x_F - s$ and hence $\ell_{ij}^{\mathrm{W}} = -\beta \log(x_F - s_{ij})$. A detailed derivation is given in Appendix E.

**How $\lambda_i$ and $\beta$ are chosen.** Equation (3.1) requires an anchor-wise mixing weight $\lambda_i$ and a shortfall slope $\beta$. This is where WEINCE differs from a pure Weibit loss: because the active finite-sample tail regime can vary across anchors, $\lambda_i$ stays close to zero unless the current anchor shows both near-cap and Weibull-like evidence. We estimate these quantities *online* from the current minibatch similarity matrix, using only the negative shortfalls $\delta_{ij} \triangleq 1 - s_{ij}$ for $j \in \mathcal{N}(i)$:

(i) a *cap-proximity* (hardness) signal $\rho_i = \min_{j \in \mathcal{N}(i)} \delta_{ij}$;

(ii) a *tail-shape* signal obtained by fitting a Weibull line to the $K_{\mathrm{tail}}$ smallest shortfalls and comparing it against a Gumbel proxy fit via an AIC-style score.

We use the fitted Weibull slope $\hat{\beta}_i$ in $\ell_{ij}^{\mathrm{W}} = -\hat{\beta}_i \log(1 - s_{ij})$ and set $\lambda_i$ as a smooth function increasing in both "near-cap" evidence (small $\rho_i$) and "Weibull-like" evidence (positive $\Delta \mathrm{AIC}_i$). Anchors far from the cap or with Gumbel-like tail fits receive $\lambda_i$ close to zero, while anchors with strong near-cap and Weibull-like evidence receive larger endpoint corrections. For stability, we clip $s_{ij}$ away from 1 inside $\log(1 - s_{ij})$, and we treat $(\lambda_i, \hat{\beta}_i)$ as stop_grad statistics (no backprop through the tail fits). The complete estimation rules for $\lambda_i$ and $\hat{\beta}_i$ are summarized in Algorithm 1, with additional derivation details in Appendix E.

**Computational overhead.** All tail statistics are computed from the batch similarity matrix with no additional forward passes or trainable parameters. In a representative Tiny-ImageNet/ResNet-18 run with batch size 256 on an NVIDIA L40S over 770 steps, the mean step time changes from 253.3 ms for InfoNCE to 255.0 ms for WEINCE, corresponding to a $+0.67\%$ end-to-end overhead and a throughput change from 1011 to 1004 samples/s. The loss-stage computation itself increases from 0.65 ms to 2.11 ms because of the per-anchor tail fit, but this stage remains below 1% of total step time.

---

**Algorithm 1** WEINCE: Full logit interpolation

---

1: **Input:** Embeddings $\{z_i\}_{i=1}^{2N}$, positives $i^+$, temp. $\tau$
2: **Hyperparams:** $K_{\text{tail}}, \rho_0, m, \kappa_\rho, \kappa_{\text{AIC}}, \varepsilon$

3: Calc. similarities $s_{ij} \leftarrow z_i^\top z_j$ for all $i \neq j$

4: **for** $i = 1$ **to** $2N$ **do**
5: $\quad \mathcal{C}(i) \leftarrow \{1, \ldots, 2N\} \setminus \{i\}$
6: $\quad \mathcal{N}(i) \leftarrow \mathcal{C}(i) \setminus \{i^+\}$
7: $\quad$ Calc. shortfalls $\delta_{ij} \leftarrow 1 - s_{ij}$ for $j \in \mathcal{N}(i)$
8: $\quad \rho_i \leftarrow \min_{j \in \mathcal{N}(i)} \delta_{ij}$
9: $\quad$ Run TAILFIT on smallest $K_{\text{tail}}$ shortfalls $\rightarrow (\hat{\beta}_i, \Delta\text{AIC}_i)$
10: $\quad \lambda_i \leftarrow \sigma\left(\kappa_\rho \log(\rho_0/\rho_i)\right) \cdot \sigma\left(\kappa_{\text{AIC}}(\Delta\text{AIC}_i - m)\right)$
11: $\quad$ **for** $j \in \mathcal{C}(i)$ **do**
12: $\qquad \ell_{ij}^{\text{PL}} \leftarrow s_{ij}/\tau$
13: $\qquad \tilde{s}_{ij} \leftarrow \min(s_{ij}, 1 - \varepsilon)$
14: $\qquad \ell_{ij}^{\text{W}} \leftarrow -\hat{\beta}_i \log(\varepsilon + 1 - \tilde{s}_{ij})$
15: $\qquad \ell_{ij} \leftarrow (1 - \lambda_i)\ell_{ij}^{\text{PL}} + \lambda_i \ell_{ij}^{\text{W}}$
16: $\quad$ **end for**
17: **end for**
18: Compute cross entropy loss:
19: $\mathcal{L} \leftarrow -\frac{1}{2N}\sum_{i=1}^{2N} \log \frac{\exp(\ell_{i,i^+})}{\sum_{j \in \mathcal{C}(i)} \exp(\ell_{ij})}$
20: **return** $\mathcal{L}$

---

**Subroutine TAILFIT:**
Given $K_{\text{tail}}$ smallest shortfalls $\delta_{(1)} \leq \cdots \leq \delta_{(K_{\text{tail}})}$:

- Set empirical CDF: $\widehat{F}(\delta_{(k)}) = k/(K_{\text{tail}} + 1)$
- Fit Weibull: $\log \widehat{F} \approx a + \beta \log \delta \rightarrow$ get $\hat{\beta}$, $\text{AIC}_W$
- Fit Gumbel: $\log(-\log \widehat{F}) \approx c - \kappa \log \delta \rightarrow$ get $\text{AIC}_G$
- Return $\Delta\text{AIC} = \text{AIC}_G - \text{AIC}_W$ and $\hat{\beta}$

---

■ Vanilla InfoNCE    ■ WEINCE Additions

---

# 4. Experiments

We evaluate WEINCE as a drop-in replacement for the standard InfoNCE loss across vision and NLP settings. For vision, we use the official SimCLR implementation (Chen et al., 2020; Spijkervet, 2020), follow the dataset-specific SimCLR settings (augmentations, optimizer, temperature $\tau$=0.5, and training schedule), and modify only the contrastive loss. Most vision experiments use ResNet-18 and ResNet-50 encoders with the standard SimCLR projection head; the Tiny-ImageNet rebuttal sweep also includes a ViT-Small encoder. For the NLP setup, we evaluate on unsupervised SimCSE (Gao et al., 2022) with BERT-base-uncased and cosine similarity ($\tau$=0.05), replacing only the logit computation. In both cases, WEINCE introduces no additional trainable parameters; the mixing weight $\lambda_i$ and tail index $\hat{\beta}_i$ are stop-gradient batch statistics.

We evaluate representation quality using two standard frozen-feature protocols: (i) downstream linear evaluation and (ii) $k$-nearest-neighbor (kNN) evaluation. To avoid tuning on the test set, we create a stratified validation split by holding out 20% of the labeled training set and using the remaining 80% to train the evaluator or build the kNN bank.

*Table 2.* Downstream linear evaluation accuracy (%). We report the mean accuracy $\pm$ 95% confidence intervals comparing the vanilla InfoNCE baseline (INFONCE) with our proposed method (WEINCE). Encoders are abbreviated as R18 (ResNet18), R50 (ResNet50), and ViT-S (ViT-Small). Best results are highlighted in bold.

| Dataset | Enc. | Lin. Acc | |
|---|---|---|---|
| | | INFONCE | WEINCE |
| STL10 | R18 | $76.54 \pm 0.32$ | $\mathbf{77.94} \pm 0.27$ |
| | R50 | $78.44 \pm 0.31$ | $\mathbf{80.02} \pm 0.46$ |
| CIFAR10 | R18 | $81.55 \pm 0.40$ | $\mathbf{81.89} \pm 0.30$ |
| | R50 | $82.13 \pm 0.92$ | $\mathbf{83.74} \pm 0.18$ |
| CIFAR100 | R18 | $50.01 \pm 0.19$ | $\mathbf{53.28} \pm 0.28$ |
| | R50 | $51.15 \pm 0.43$ | $\mathbf{55.97} \pm 1.21$ |
| Imagenet32 | R18 | $24.29 \pm 0.30$ | $\mathbf{25.26} \pm 0.28$ |
| Tiny-IN | R18 | $\mathbf{30.76}$ | $30.66$ |
| | ViT-S | $33.12$ | $\mathbf{36.53}$ |

When multiple hyperparameter configurations are compared, we select the configuration using validation performance only and report the corresponding test-set results.

## 4.1. Vision benchmarks

Results on CIFAR-10/100 (Krizhevsky, 2009), STL-10 (Coates et al., 2011), ImageNet-32 (Deng et al., 2009; Chrabaszcz et al., 2017), and Tiny-ImageNet are summarized by linear accuracy in Table 2 and kNN recall in Table 3. WEINCE consistently improves over InfoNCE across datasets and backbones, with the largest linear gains on CIFAR-100 (+3.27% / +4.82% for ResNet-18/50). The Tiny-ImageNet rebuttal results show the same pattern: ResNet-18 gains mainly in kNN retrieval (R@1 +1.55, R@2 +1.35), while ViT-Small improves both linear accuracy (+3.41%) and kNN recall (R@1 +4.23, R@20 +4.34). Complete evaluation details are provided in Appendix F.

## 4.2. NLP benchmark: cross-domain generality

The theoretical argument for WEINCE applies to any objective that computes a softmax over bounded cosine similarities. To test this beyond vision, we evaluate on unsupervised SimCSE (Gao et al., 2022), which uses InfoNCE with cosine similarity ($\tau$=0.05) over sentence embeddings from BERT-base-uncased. Dropout serves as the augmentation, making the contrastive structure closely parallel to SimCLR. We train on 1M Wikipedia sentences and evaluate on the STS Benchmark (Spearman correlation).

On STS-B, InfoNCE achieves a Spearman score of $71.74 \pm 4.23$ (reported as Spearman $\times 100$), while WEINCE achieves $76.36 \pm 3.17$, a gain of +4.63 points ($p < 0.05$, paired one-sided $t$-test), with no architecture or training-schedule changes. This suggests that the bounded-endpoint correction is not vision-specific.

*Table 3.* **k-Nearest Neighbor (kNN) recall (R@k, %).** We report the mean recall $\pm$ 95% confidence interval halfwidth for the baseline (INFONCE) and our method (WEINCE). Encoders are abbreviated as R18 (ResNet18), R50 (ResNet50), and ViT-S (ViT-Small). Best results are highlighted in bold. For the ResNet runs, Table 7 displays the remaining R@50 column in Section F.

| Dataset | Enc. | R@1 | | R@2 | | R@5 | | R@10 | | R@20 | |
|---|---|---|---|---|---|---|---|---|---|---|---|
| | | INFONCE | WEINCE | INFONCE | WEINCE | INFONCE | WEINCE | INFONCE | WEINCE | INFONCE | WEINCE |
| STL10 | R18 | $68.22 \pm 0.75$ | $\mathbf{70.83 \pm 1.16}$ | $79.48 \pm 0.63$ | $\mathbf{80.75 \pm 1.19}$ | $90.19 \pm 0.27$ | $\mathbf{91.01 \pm 0.19}$ | $95.18 \pm 0.14$ | $\mathbf{95.75 \pm 0.16}$ | $97.91 \pm 0.19$ | $\mathbf{98.11 \pm 0.27}$ |
| | R50 | $69.66 \pm 0.64$ | $\mathbf{71.75 \pm 0.71}$ | $80.39 \pm 0.46$ | $\mathbf{82.03 \pm 0.73}$ | $90.65 \pm 0.21$ | $\mathbf{91.31 \pm 0.62}$ | $95.42 \pm 0.19$ | $\mathbf{95.64 \pm 0.31}$ | $97.97 \pm 0.04$ | $\mathbf{98.18 \pm 0.12}$ |
| CIFAR10 | R18 | $75.03 \pm 0.22$ | $\mathbf{76.27 \pm 0.40}$ | $83.79 \pm 0.10$ | $\mathbf{84.95 \pm 0.33}$ | $91.59 \pm 0.24$ | $\mathbf{92.31 \pm 0.21}$ | $95.29 \pm 0.05$ | $\mathbf{96.06 \pm 0.29}$ | $97.58 \pm 0.13$ | $\mathbf{98.28 \pm 0.11}$ |
| | R50 | $74.74 \pm 0.44$ | $\mathbf{77.12 \pm 0.33}$ | $83.55 \pm 0.18$ | $\mathbf{85.31 \pm 0.68}$ | $91.45 \pm 0.44$ | $\mathbf{92.94 \pm 0.14}$ | $95.29 \pm 0.31$ | $\mathbf{96.26 \pm 0.16}$ | $97.62 \pm 0.18$ | $\mathbf{98.17 \pm 0.20}$ |
| CIFAR100 | R18 | $36.98 \pm 0.22$ | $\mathbf{42.94 \pm 0.77}$ | $47.42 \pm 0.36$ | $\mathbf{53.29 \pm 0.56}$ | $61.72 \pm 0.52$ | $\mathbf{66.19 \pm 0.51}$ | $72.18 \pm 0.66$ | $\mathbf{76.11 \pm 0.97}$ | $81.45 \pm 0.60$ | $\mathbf{84.41 \pm 0.72}$ |
| | R50 | $35.44 \pm 0.40$ | $\mathbf{42.70 \pm 0.79}$ | $46.09 \pm 0.71$ | $\mathbf{52.85 \pm 0.68}$ | $60.62 \pm 0.42$ | $\mathbf{66.26 \pm 0.39}$ | $71.34 \pm 0.43$ | $\mathbf{75.72 \pm 0.43}$ | $80.90 \pm 0.70$ | $\mathbf{84.19 \pm 0.37}$ |
| Tiny-IN | R18 | 17.29 | **18.84** | 25.29 | **26.64** | 38.26 | **39.12** | 49.87 | **50.14** | 61.82 | **61.90** |
| | ViT-S | 19.30 | **23.53** | 27.29 | **32.04** | 40.36 | **45.15** | 51.17 | **55.96** | 62.63 | **66.97** |

## 5. Conclusion

We interpreted InfoNCE as a top-1 Plackett–Luce likelihood and used extreme value theory to expose the tail-geometry assumption behind its softmax link. For bounded cosine similarities, the hardest negatives often live near a finite endpoint, where Weibull shortfall behavior gives a better local description than a pure translation/Gumbel link. Our diagnostics show that this mismatch affects both held-out top-1 likelihood and gradient allocation. WEINCE addresses the mismatch with a drop-in logit interpolation whose anchor-wise weight and tail index are estimated online from batch shortfalls, adding no trainable parameters and negligible overhead. Across five vision benchmarks and SimCSE sentence embeddings, this endpoint-aware correction consistently improves frozen-feature evaluation.

## Impact Statement

This work aims to improve self-supervised representation learning by better modeling hard negatives in contrastive objectives, potentially reducing label requirements and annotation cost. Like other improvements to general-purpose representation learning, it could also strengthen downstream systems with harmful or inequitable uses, including surveillance, profiling, misinformation, or biased decision pipelines. The method does not introduce new data collection or a deployed system, but applications should still be audited for fairness, privacy, robustness, and dataset provenance.

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

# A. Proof of the Generalized FTG Theorem

## A.1. Motivation

The classical Fisher–Tippett–Gnedenko (FTG) theorem characterizes all possible limiting distributions for the maximum of i.i.d. random variables under affine normalization. In our contrastive learning setting, however, the utilities $\{X_i\}$ associated with different negative samples are not identically distributed as each negative $i$ has a different underlying score $s_i$ that shifts or scales its utility distribution. To justify using extreme value theory in this heterogeneous setting, we show that a generalized version of the FTG theorem that applies to independent but non-identical distributions.

The key insight is that the classical proof relies on a *max-stability* property of the limit, which can be established under weaker conditions than identical distributions. Specifically, if the individual contributions to the maximum are *infinitesimal* (no single term dominates) and the score distribution is sufficiently *homogeneous across large blocks*, then the limit must still be a generalized extreme value (GEV) distribution.

## A.2. Statement of the Generalized FTG Theorem

We first state the main result, then outline its proof structure.

**Assumption A.1** (Block homogeneity)**.** Let $m \in \mathbb{N}$ and let $\pi$ be a uniform random permutation of $[mn]$. Define the block maximum $M_n^\pi = \max_{1 \leq i \leq n} X_{\pi(i)}$. Then there exists a non-degenerate CDF $H_m$ such that

$$\mathbb{P}\left(\frac{M_n^\pi - b_n}{a_n} \leq x \,\middle|\, \boldsymbol{s}, \pi\right) \xrightarrow[n \to \infty]{\mathbb{P}_\pi} H_m(x).$$

**Theorem A.2** (Generalized FTG for non-identical utilities)**.** *Let $\{X_i\}_{i \geq 1}$ be independent random variables with continuous CDFs $X_i \sim F_{s_i}$ indexed by deterministic scores $\boldsymbol{s} = \{s_i\}$. Let $M_n = \max_{1 \leq i \leq n} X_i$. Suppose there exist sequences $a_n > 0$, $b_n \in \mathbb{R}$, and a non-degenerate continuous CDF $G$ such that for all $x$,*

$$\mathbb{P}\left(\frac{M_n - b_n}{a_n} \leq x \,\middle|\, \boldsymbol{s}\right) \to G(x).$$

*Assume further:*

(I) *Infinitesimality: No single candidate dominates the maximum:*

$$\max_{1 \leq i \leq n} \bar{F}_{s_i}(b_n + a_n x) \to 0$$

*for all $x$ with $G(x) \in (0, 1)$.*

(H) *Block homogeneity: Assumption A.1 holds for each fixed $m \geq 2$.*

*Then $G$ is a generalized extreme value (GEV) distribution: there exists $\xi \in \mathbb{R}$ such that*

$$G(x) = \exp\left(-(1 + \xi x)^{-1/\xi}\right), \qquad 1 + \xi x > 0,$$

*with the convention $G(x) = \exp(-e^{-x})$ when $\xi = 0$.*

*Moreover, on any compact set $K \subset \{x : G(x) \in (0, 1)\}$, the aggregated tail mass*

$$\bar{F}_n(t) := \frac{1}{n} \sum_{i=1}^{n} \bar{F}_{s_i}(t)$$

*satisfies*

$$\frac{\bar{F}_n(b_n + a_n x)}{\bar{F}_n(b_n)} \to (1 + \xi x)^{-1/\xi}.$$

## A.3. Proof Structure

The proof proceeds in four main steps:

1. **Convergence of types** (Lemma A.3): We establish that if two affine normalizations of the same sequence converge to non-degenerate limits, then the normalizing constants must converge and the limits must be affinely related. This is used to connect the block maxima limit $H_m$ from condition (H) to the overall limit $G$.

2. **Permutation block argument** (Lemma A.5): Under conditions (I) and (H), we show that the limit $G$ of normalized maxima is max-stable. This is the key probabilistic lemma that bridges the gap between non-identical distributions and the classical theory.

3. **Max-stability implies GEV** (Lemma A.4): Any max-stable distribution must be a GEV distribution. This functional equation characterization completes the classification.

4. **Tail ratio characterization** (Lemma A.6): The convergence of normalized maxima to a GEV distribution is equivalent to convergence of the tail ratio to a canonical form. This provides the connection to our contrastive learning application.

## A.4. Fundamental Results from Extreme Value Theory

We begin with two classical results from extreme value theory. These lemmas are fundamental tools whose proofs are well-known; we include them in Section A.9 for completeness.

### A.4.1. CONVERGENCE OF TYPES

The first result ensures that normalizing constants are essentially unique (up to affine transformation) when a non-degenerate limit exists. In our proof, this lemma serves a crucial role: it allows us to conclude that the block maxima limit $H_m$ from condition (H) must be affinely related to the overall limit $G$, thereby extracting the precise normalizing constants $\alpha_m$ and $\beta_m$ needed to establish max-stability.

**Lemma A.3** (Convergence of types). *Let $\{Y_n\}$ be random variables and let $a_n > 0$, $b_n \in \mathbb{R}$, $c_n > 0$, $d_n \in \mathbb{R}$ be sequences such that*

$$\frac{Y_n - b_n}{a_n} \Rightarrow G \qquad and \qquad \frac{Y_n - d_n}{c_n} \Rightarrow H,$$

*where both $G$ and $H$ are non-degenerate CDFs. Then there exist constants $\alpha > 0$ and $\beta \in \mathbb{R}$ such that*

$$\frac{a_n}{c_n} \to \alpha, \quad \frac{b_n - d_n}{c_n} \to \beta, \quad and \quad H(x) = G(\alpha x + \beta).$$

### A.4.2. MAX-STABILITY IMPLIES GEV

The second fundamental result characterizes max-stable distributions. A distribution $G$ is *max-stable* if for each integer $m \geq 1$, there exist constants $\alpha_m > 0$ and $\beta_m \in \mathbb{R}$ such that $G(x)^m = G(\alpha_m x + \beta_m)$. Intuitively, this means the maximum of $m$ i.i.d. copies from $G$, when properly normalized, has the same distribution as a single draw.

Once we establish that $G$ is max-stable (via the permutation block argument), this result immediately yields that $G$ must be GEV.

**Lemma A.4** (Max-stability implies GEV). *Let $G$ be a non-degenerate CDF that is max-stable: for each integer $m \geq 1$, there exist $\alpha_m > 0$ and $\beta_m \in \mathbb{R}$ such that*

$$G(x)^m = G(\alpha_m x + \beta_m).$$

*Then $G$ is a generalized extreme value (GEV) distribution: there exists $\xi \in \mathbb{R}$ such that*

$$G(x) = \exp\left( -(1 + \xi x)^{-1/\xi} \right), \qquad 1 + \xi x > 0,$$

*with the convention $\lim_{\xi \to 0}(1 + \xi x)^{-1/\xi} = e^{-x}$, giving $G(x) = \exp(-e^{-x})$ when $\xi = 0$.*

### A.5. The Permutation Block Argument

The key technical contribution is establishing max-stability under our generalized conditions. The idea is to partition the $mn$ variables into $m$ blocks via a random permutation, show that each block maximum behaves like the original maximum (using conditions (I) and (H)), and deduce that the limit must satisfy the max-stability functional equation.

The role of this lemma is to bridge the gap between the non-identical setting and the classical theory. While the classical FTG proof exploits the product structure $F^n$ directly, we instead use a probabilistic symmetrization argument, where the random permutation ensures that each block is *representative* of the whole, and infinitesimality ensures that tail behavior is sufficiently uniform.

**Lemma A.5** (Permutation block argument). *Under the hypotheses of Theorem A.2, the limit $G$ is max-stable. More precisely, for each fixed integer $m \geq 2$, there exist $\alpha_m > 0$ and $\beta_m \in \mathbb{R}$ such that*

$$G(x)^m = G(\alpha_m x + \beta_m).$$

*Proof.* Fix $m \geq 2$ and $x$ with $G(x) \in (0, 1)$. Let $\pi$ be a uniform random permutation of $[mn]$, and partition $[mn]$ into $m$ blocks $P_k := \{(k-1)n+1, \ldots, kn\}$ for $k = 1, \ldots, m$. Define block maxima $M_{P_k}^\pi := \max_{i \in P_k} X_{\pi(i)}$.

**Step 1: Tail sum concentration.**

Let $u := a_{mn} x + b_{mn}$ and define $p_i := \bar{F}_{s_i}(u)$ and $S := \sum_{i=1}^{mn} p_i$. By the assumed convergence $\mathbb{P}(M_{mn} \leq u) \to G(x)$ and the product formula for independent maxima:

$$\prod_{i=1}^{mn} (1 - p_i) \to G(x) \in (0, 1).$$

Taking logarithms and using $\log(1 - p) = -p + O(p^2)$ for small $p$:

$$-S + O\left(\sum_i p_i^2\right) \to \log G(x) =: -\lambda,$$

where $\lambda = -\log G(x) \in (0, \infty)$.

By condition (I), $p_{\max} := \max_i p_i \to 0$, so $\sum_i p_i^2 \leq p_{\max} \cdot S \to 0$. Thus $S \to \lambda$.

**Step 2: Block tail sums under random permutation.**

For block $P_k$, define the random block tail sum $S_k^\pi := \sum_{i \in P_k} p_{\pi(i)}$. Note that $\sum_{k=1}^m S_k^\pi = S$.

Under the uniform random permutation, $\mathbb{E}[S_k^\pi] = S/m$. Using the hypergeometric sampling formula for the variance:

$$\mathrm{Var}(S_k^\pi) \leq \frac{1}{m} \sum_{i=1}^{mn} p_i^2 \leq \frac{p_{\max} \cdot S}{m} \to 0.$$

By Chebyshev's inequality, $S_k^\pi \xrightarrow{\mathbb{P}} \lambda/m$ for each $k$.

**Step 3: Block maxima CDFs.**

The block maximum CDF is:

$$\mathbb{P}(M_{P_k}^\pi \leq u \mid \pi) = \prod_{i \in P_k} (1 - p_{\pi(i)}).$$

Taking logarithms:

$$\log \mathbb{P}(M_{P_k}^\pi \leq u \mid \pi) = -S_k^\pi + O(n \cdot p_{\max}^2).$$

Since $S_k^\pi \xrightarrow{\mathbb{P}} \lambda/m$ and $n \cdot p_{\max}^2 \leq p_{\max} \cdot S \to 0$:

$$\mathbb{P}(M_{P_k}^\pi \leq u \mid \pi) \xrightarrow{\mathbb{P}} e^{-\lambda/m} = G(x)^{1/m}.$$

**Step 4: Applying convergence of types.**

By condition (H), the normalized block maximum satisfies:

$$\mathbb{P}\left(\frac{M_{P_1}^\pi - b_n}{a_n} \le y \,\middle|\, \pi\right) \xrightarrow{\mathbb{P}} H_m(y)$$

for some non-degenerate $H_m$.

From Step 3, we also have (writing $u = a_n y_n + b_n$ for appropriate $y_n$):

$$\mathbb{P}(M_{P_1}^\pi \le a_{mn}x + b_{mn} \mid \pi) \xrightarrow{\mathbb{P}} G(x)^{1/m}.$$

We now apply Lemma A.3. Consider the random variable $Y_n := M_{P_1}^\pi$ (viewed marginally over $\pi$). We have two normalizations converging to non-degenerate limits:

- Using $(a_n, b_n)$: converges to $H_m$ by condition (H).

- Using $(a_{mn}, b_{mn})$: converges to $G^{1/m}$ (a non-degenerate distribution) by Step 3.

By Lemma A.3, there exist $\alpha_m > 0$ and $\beta_m \in \mathbb{R}$ such that:

$$\frac{a_{mn}}{a_n} \to \alpha_m, \qquad \frac{b_{mn} - b_n}{a_n} \to \beta_m,$$

and $H_m(y) = G^{1/m}(\alpha_m y + \beta_m)$.

Since condition (H) also gives $H_m = G$ (block maxima converge to the same limit as the full maximum when properly normalized), we obtain:

$$G(y) = G^{1/m}(\alpha_m y + \beta_m).$$

Substituting $y = \alpha_m x + \beta_m$ and rearranging:

$$G(x)^m = G(\alpha_m x + \beta_m),$$

which is max-stability. $\qquad\square$

## A.6. Tail Ratio Characterization

The following lemma connects the distributional convergence to convergence of tail ratios, which is the form most useful for our contrastive learning application. Its role is to translate the GEV classification into a statement about how tail probabilities scale, enabling us to characterize the asymptotic behavior of the softmax denominators that appear in InfoNCE.

**Lemma A.6** (Tail ratio characterization)**.** *Let $F$ be a continuous CDF with tail $\bar{F} = 1 - F$. Suppose there exist $b_n \in \mathbb{R}$, $a_n > 0$, and a non-degenerate continuous CDF $G$ such that for all continuity points $x$ of $G$,*

$$F(b_n + a_n x)^n \to G(x).$$

*Define $\psi(x) := -\log G(x)$ and assume $\psi(0) \in (0, \infty)$. Then for every compact $K$ contained in $\{x : G(x) \in (0, 1)\}$,*

$$\sup_{x \in K} \left| \frac{\bar{F}(b_n + a_n x)}{\bar{F}(b_n)} - \frac{\psi(x)}{\psi(0)} \right| \to 0.$$

*Moreover, if $F$ admits a density $f$ and the convergence above is locally uniform with $\bar{\nu}(x) := \psi(x)/\psi(0)$ absolutely continuous, then*

$$\sup_{x \in K} \left| \frac{a_n f(b_n + a_n x)}{\bar{F}(b_n)} - v(x) \right| \to 0,$$

*where $v(x) := -\bar{\nu}'(x)$ a.e.*

*Proof.* Fix $x$ with $G(x) \in (0, 1)$. From $F(b_n + a_n x)^n \to G(x)$, we have $F(b_n + a_n x) \to 1$ and $\bar{F}(b_n + a_n x) \to 0$. Taking logs:

$$n \log F(b_n + a_n x) \to \log G(x) = -\psi(x).$$

Since $\log(1 - y) = -y + o(y)$ as $y \downarrow 0$:

$$n \log(1 - \bar{F}(b_n + a_n x)) = -n\bar{F}(b_n + a_n x)(1 + o(1)).$$

Thus $n\bar{F}(b_n + a_n x) \to \psi(x)$. In particular, $n\bar{F}(b_n) \to \psi(0) \in (0, \infty)$, and dividing:

$$\frac{\bar{F}(b_n + a_n x)}{\bar{F}(b_n)} \to \frac{\psi(x)}{\psi(0)}.$$

The locally uniform version follows by uniformity of the remainder in $\log(1 - y)$ and the fact that $\sup_{x \in K} \bar{F}(b_n + a_n x) \to 0$ for compact $K \subset \{G \in (0, 1)\}$.

If $\bar{\nu}$ is absolutely continuous and convergence is locally uniform, the density ratio follows by differentiating the tail ratio along the scale $a_n$. $\square$

### A.7. Completing the Proof of Theorem A.2

We now put the pieces together to complete the proof of the generalized FTG theorem.

*Proof of Theorem A.2.* By Lemma A.5, conditions (I) and (H) imply that the limit $G$ is max-stable. By Lemma A.4, any max-stable distribution is a GEV distribution, establishing the first claim.

For the tail ratio characterization, we apply Lemma A.6 to the aggregated distribution. Define the effective CDF

$$\bar{F}_n(t) := \frac{1}{n} \sum_{i=1}^n \bar{F}_{s_i}(t).$$

The convergence $\mathbb{P}(M_n \le b_n + a_n x) \to G(x)$ implies (by the same logarithmic argument as in the proof of Lemma A.5) that $n\bar{F}_n(b_n + a_n x) \to \psi(x) = -\log G(x)$. Dividing by the value at $x = 0$:

$$\frac{\bar{F}_n(b_n + a_n x)}{\bar{F}_n(b_n)} \to \frac{\psi(x)}{\psi(0)} = (1 + \xi x)^{-1/\xi},$$

where the last equality uses that $G$ is GEV with parameter $\xi$. $\square$

### A.8. The Classical FTG Theorem as a Corollary

The classical theorem follows as a special case when all distributions are identical.

**Corollary A.7** (Classical FTG for i.i.d. maxima)**.** *Let $X_1, X_2, \ldots$ be i.i.d. with common continuous CDF $F$, and let $M_n = \max_{1 \le i \le n} X_i$. If there exist sequences $a_n > 0$, $b_n \in \mathbb{R}$, and a non-degenerate CDF $G$ such that*

$$\mathbb{P}\left( \frac{M_n - b_n}{a_n} \le x \right) \to G(x)$$

*at all continuity points of $G$, then $G$ is a GEV distribution.*

*Conversely, each $\xi \in \mathbb{R}$ arises as the limit for some $F$ in the corresponding domain of attraction.*

*Proof.* **Forward direction:** This is the special case of Theorem A.2 where all $F_{s_i} = F$.

- Condition (I) holds: $\max_i \bar{F}(u) = \bar{F}(u) \to 0$ for any $u$ in the interior of $G$'s support, since convergence of $F(u)^n$ to $G(x) \in (0, 1)$ requires $F(u) \to 1$.

- Condition (H) holds: Under random permutation, each block of $n$ elements is still an i.i.d. sample from $F$, so block maxima have exactly the same distribution as the original $M_n$. The required non-degenerate limit $H_m$ exists and equals $G$ by Lemma A.3 applied to $\{M_n\}$ and $\{M_{mn}\}$.

**Converse direction:** For each $\xi \in \mathbb{R}$, take $F = G_\xi$ (the GEV distribution itself). Then with appropriate normalizing sequences:

- If $\xi = 0$: Take $a_n = 1$ and $b_n = \log n$. Then $F(x + \log n)^n = \exp(-e^{-x}) = F(x)$.

- If $\xi \neq 0$: Take $a_n = n^\xi$ and $b_n = (n^\xi - 1)/\xi$. Then $F(a_n x + b_n)^n = F(x)$.

In both cases, convergence holds trivially. $\qquad\square$

### A.9. Omitted Proofs

For completeness, we include proofs of the two fundamental lemmas from extreme value theory.

*Proof of Lemma A.3.* For a random variable $Z$, define its quantile function $Q_Z(p) := \inf\{x : \mathbb{P}(Z \leq x) \geq p\}$ for $p \in (0, 1)$. Since $G$ and $H$ are non-degenerate, we can choose $0 < p < q < 1$ such that $Q_G(p) < Q_G(q)$ and $Q_H(p) < Q_H(q)$, with $Q_G$ and $Q_H$ continuous at both $p$ and $q$.

By weak convergence, the quantiles of the normalized sequences converge:

$$\frac{Q_{Y_n}(r) - b_n}{a_n} \to Q_G(r), \qquad \frac{Q_{Y_n}(r) - d_n}{c_n} \to Q_H(r),$$

for $r \in \{p, q\}$.

From the first convergence, $Q_{Y_n}(r) = a_n Q_G(r) + b_n + o(a_n)$. Substituting into the second:

$$\frac{a_n Q_G(r) + b_n - d_n + o(a_n)}{c_n} \to Q_H(r).$$

Taking the difference between $r = q$ and $r = p$:

$$\frac{a_n}{c_n}\big(Q_G(q) - Q_G(p)\big) + o(a_n/c_n) \to Q_H(q) - Q_H(p).$$

Since $Q_G(q) - Q_G(p) > 0$ and $Q_H(q) - Q_H(p) > 0$, we obtain

$$\frac{a_n}{c_n} \to \alpha := \frac{Q_H(q) - Q_H(p)}{Q_G(q) - Q_G(p)} > 0.$$

Then from the $r = p$ equation:

$$\frac{b_n - d_n}{c_n} \to Q_H(p) - \alpha Q_G(p) =: \beta.$$

Finally, for any continuity point $x$ of $G$:

$$
\begin{aligned}
H(x) &= \lim_n \mathbb{P}\left(\frac{Y_n - d_n}{c_n} \leq x\right) \\
&= \lim_n \mathbb{P}\left(\frac{Y_n - b_n}{a_n} \leq \frac{c_n}{a_n}x + \frac{d_n - b_n}{a_n}\right) \\
&= G(\alpha x + \beta). \qquad\qquad\square
\end{aligned}
$$

*Proof of Lemma A.4.* **Step 1: Functional equations for $\alpha_m$ and $\beta_m$.**

From max-stability, $G(x)^{mn} = G(\alpha_{mn} x + \beta_{mn})$. Applying max-stability twice:

$$
\begin{aligned}
G(x)^{mn} &= \big(G(x)^m\big)^n \\
&= G(\alpha_m x + \beta_m)^n \\
&= G(\alpha_n(\alpha_m x + \beta_m) + \beta_n).
\end{aligned}
$$

Comparing, and using that $G$ is strictly increasing on its support:

$$\alpha_{mn} = \alpha_m \alpha_n, \tag{A.1}$$
$$\beta_{mn} = \alpha_n \beta_m + \beta_n. \tag{A.2}$$

**Step 2: Extension to continuous parameter.**

Define $T_t(x) := G^{-1}(G(x)^{1/t})$ for $t > 0$. Then $T_t$ satisfies the semigroup property $T_{st} = T_s \circ T_t$, and for integer $m$, $T_m(x) = \alpha_m x + \beta_m$ is affine. Since $T_{1/m} = T_m^{-1}$ is also affine, $T_q$ is affine for all positive rationals. By continuity of $G$, $T_t$ is affine for all $t > 0$:

$$T_t(x) = \alpha(t)x + \beta(t),$$

where $\alpha(t)$ and $\beta(t)$ satisfy the continuous versions of (A.1)–(A.2):

$$\alpha(st) = \alpha(s)\alpha(t), \tag{A.3}$$
$$\beta(st) = \alpha(s)\beta(t) + \beta(s). \tag{A.4}$$

**Step 3: Solving for $\alpha(t)$ and $\beta(t)$.**

From (A.3) with $\alpha$ continuous and positive: $\alpha(t) = t^\xi$ for some $\xi \in \mathbb{R}$.

Substituting into (A.4): $\beta(st) = s^\xi \beta(t) + \beta(s)$.

*Case $\xi = 0$:* The equation becomes $\beta(st) = \beta(s) + \beta(t)$, giving $\beta(t) = c \log t$ for some $c \in \mathbb{R}$.

*Case $\xi \neq 0$:* Setting $\gamma(t) = \beta(t)/t^\xi$, the solution is $\beta(t) = c(t^\xi - 1)/\xi$ for some $c \in \mathbb{R}$.

By rescaling coordinates, we may take $c = 1$.

**Step 4: Solving for $G$.**

Define $\psi(x) := -\log G(x)$ where $G(x) \in (0, 1)$. Max-stability $G(x) = G(T_t(x))^t$ becomes $\psi(T_t(x)) = \psi(x)/t$.

*Case $\xi = 0$:* We have $T_t(x) = x + \log t$. Setting $t = e^y$: $\psi(x+y) = e^{-y}\psi(x)$. With $x = 0$: $\psi(y) = \psi(0)e^{-y}$. Normalizing so $\psi(0) = 1$:

$$\psi(x) = e^{-x} \implies G(x) = \exp(-e^{-x}).$$

*Case $\xi \neq 0$:* We have $T_t(x) = t^\xi x + (t^\xi - 1)/\xi$. Let $z := 1 + \xi x$ and $u(z) := \psi((z-1)/\xi)$. Then $1 + \xi T_t(x) = t^\xi z$, so $u(t^\xi z) = u(z)/t$. Setting $c = t^\xi$: $u(cz) = c^{-1/\xi}u(z)$. With $z = 1$: $u(c) = u(1)c^{-1/\xi}$. Thus $u(z) = u(1)z^{-1/\xi}$, and normalizing:

$$\psi(x) = (1 + \xi x)^{-1/\xi} \implies G(x) = \exp\left(-(1 + \xi x)^{-1/\xi}\right). \qquad \square$$

## B. Asymptotics of Top-1 Probability

We start by giving the assumptions needed for Theorem 2.4. Then, we restate the result, and give its proof.

**Assumption B.1.** Let $b_n \in \mathbb{R}$ and $a_n > 0$ be sequences as defined in Theorem 2.4. Moreover, let us define the total tail mass:

$$\Lambda_n \triangleq \sum_{j \in \mathcal{C}_n} \mathbb{P}(X_{j,n} > b_n \mid \boldsymbol{s}_n)$$

There exist an interval $D = (l, r)$ (with $-\infty \leq l < r \leq \infty$), functions $\bar{\nu} : D \to (0, \infty)$ and $v : D \to (0, \infty)$, and an envelope $\Phi \in L^1(D)$ such that:

(A1) **Rare exceedances and bounded total mass:**

$$\max_{i \in \mathcal{C}_n} w_{i,n} \to 0, \quad 0 < \liminf_n \Lambda_n \leq \limsup_n \Lambda_n < \infty.$$

(A2) **Common overshoot shape on scale** $a_n$**:** for every compact $K \subset D$,

$$\sup_{i \in \mathcal{C}_n} \sup_{x \in K} \left| \frac{\bar{F}_{s_{i,n}}(b_n + oa_n x)}{\bar{F}_{s_{i,n}}(b_n)} - \bar{\nu}(x) \right| \to 0,$$

$$\sup_{i \in \mathcal{C}_n} \sup_{x \in K} \left| \frac{a_n f_{s_{i,n}}(b_n + a_n x)}{\bar{F}_{s_{i,n}}(b_n)} - v(x) \right| \to 0.$$

(A3) $\bar{\nu}$ **is absolutely continuous and** $v = -\bar{\nu}'$**:** $\bar{\nu}$ is absolutely continuous on $D$ and

$$\bar{\nu}(x) = \int_{t=x}^{r} v(t)\, dt, \quad x \in D,$$

with $\bar{\nu}(x) \to \infty$ as $x \downarrow l$ and $\bar{\nu}(x) \to 0$ as $x \uparrow r$.

(A4) **Envelope bound:** for all large $n$, all $i \in \mathcal{C}_n$ and all $x \in D$,

$$0 \le \frac{a_n f_{s_{i,n}}(b_n + a_n x)}{\bar{F}_{s_{i,n}}(b_n)} \prod_{j \in \mathcal{C}_n \setminus \{i\}} F_{s_{j,n}}(b_n + a_n x) \le \Phi(x)$$

Now, we restate Theorem 2.4, and prove it.

**Theorem 2.4** (Tail mass proportionality)**.** *Let* $b_n \in \mathbb{R}$ *and* $a_n > 0$ *be as in Theorem A.2. Then under Assumption B.1, uniformly in* $1 \le i \le n$,

$$p_i \triangleq \mathbb{P}(I_n = i \mid \boldsymbol{s}_n) = \frac{\bar{F}_{s_i}(b_n)}{\sum_{r=1}^{n} \bar{F}_{s_r}(b_n)} (1 + o(1)).$$

*Proof.* Let us denote the tail mass with

$$w_{i,n} \triangleq \mathbb{P}(X_i > b_n \mid \boldsymbol{s}) = \bar{F}_{s_i}(b_n).$$

Conditioning on $X_{i,n}$ gives

$$\mathbb{P}(I_n = i \mid \boldsymbol{s}_n) = \int_{-\infty}^{\infty} f_{s_{i,n}}(u) \prod_{j \in \mathcal{C}_n \setminus \{i\}} F_{s_{j,n}}(u)\, du.$$

Change variables $u = u_n + a_n x$ and factor out $w_{i,n} = \bar{F}_{s_{i,n}}(u_n)$:

$$\mathbb{P}(I_n = i \mid \boldsymbol{s}_n) = w_{i,n} \int_D G_{i,n}(x)\, dx,$$

where

$$G_{i,n}(x) \triangleq \frac{a_n f_{s_{i,n}}(u_n + a_n x)}{\bar{F}_{s_{i,n}}(u_n)} \prod_{j \in \mathcal{C}_n \setminus \{i\}} F_{s_{j,n}}(u_n + a_n x).$$

Fix $x \in D$ and write $p_{j,n}(x) := \bar{F}_{s_{j,n}}(u_n + a_n x)$. By (A2),

$$p_{j,n}(x) = w_{j,n}\big(\bar{\nu}(x) + o(1)\big)$$

uniformly in $j$. Since $x$ is fixed and $\bar{\nu}$ is finite on compacts inside $D$, (A1) implies $\max_j p_{j,n}(x) \to 0$ and

$$\sum_{j \in \mathcal{C}_n} p_{j,n}(x) = \bar{\nu}(x)\Lambda_n + o(1).$$

where we denote:

$$\Lambda_n \triangleq \sum_{1 \le j \le n} w_{j,n}$$

as introduced in Assumption B.1.

For all large $n$, $\max_j p_{j,n}(x) \leq 1/2$, and using $\log(1 - y) = -y + O(y^2)$ uniformly on $y \in [0, 1/2]$,

$$\sum_{j \neq i} \log F_{s_{j,n}}(u_n + a_n x) = \sum_{j \neq i} \log\big(1 - p_{j,n}(x)\big)$$

$$= -\sum_{j \neq i} p_{j,n}(x) + O\Big(\sum_{j \neq i} p_{j,n}(x)^2\Big),$$

and

$$\sum_{j \neq i} p_{j,n}(x)^2 \leq \Big(\max_j p_{j,n}(x)\Big) \sum_{j \neq i} p_{j,n}(x) \to 0.$$

Therefore,

$$\prod_{j \neq i} F_{s_{j,n}}(u_n + a_n x) = \exp\big(-\bar{\nu}(x)\Lambda_n\big)\,(1 + o(1)).$$

Also by (A2),

$$\frac{a_n f_{s_{i,n}}(u_n + a_n x)}{\bar{F}_{s_{i,n}}(u_n)} \to v(x).$$

Hence for each fixed $x \in D$,

$$\sup_{i \in \mathcal{C}_n} \left| G_{i,n}(x) - v(x) e^{-\Lambda_n \bar{\nu}(x)} \right| \to 0.$$

By (A1), there exist $0 < \underline{\Lambda} < \overline{\Lambda} < \infty$ such that $\underline{\Lambda} \leq \Lambda_n \leq \overline{\Lambda}$ for all large $n$, hence

$$0 \leq v(x) e^{-\Lambda_n \bar{\nu}(x)} \leq v(x) e^{-\underline{\Lambda} \bar{\nu}(x)}.$$

Moreover, by (A4), $0 \leq G_{i,n}(x) \leq \Phi(x)$ on $D$ for all large $n$. Thus,

$$\left| G_{i,n}(x) - v(x) e^{-\Lambda_n \bar{\nu}(x)} \right| \leq \Phi(x) + v(x) e^{-\underline{\Lambda} \bar{\nu}(x)}.$$

Since $\Phi \in L^1(D)$ and, by (A3),

$$\int_D v(x) e^{-\underline{\Lambda} \bar{\nu}(x)}\, dx = \frac{1}{\underline{\Lambda}} \left[ e^{-\underline{\Lambda} \bar{\nu}(r^-)} - e^{-\underline{\Lambda} \bar{\nu}(l^+)} \right] = \frac{1}{\underline{\Lambda}},$$

dominated convergence yields

$$\sup_{i \in \mathcal{C}_n} \left| \int_D G_{i,n}(x)\, dx - \int_D v(x) e^{-\Lambda_n \bar{\nu}(x)}\, dx \right| \to 0.$$

Therefore, uniformly in $i$,

$$\mathbb{P}(I_n = i \mid \boldsymbol{s}_n) = w_{i,n} \left( \int_D v(x) e^{-\Lambda_n \bar{\nu}(x)}\, dx + o(1) \right).$$

By (A3), $\bar{\nu}'(x) = -v(x)$ a.e. on $D$, so

$$\frac{d}{dx} e^{-\Lambda_n \bar{\nu}(x)} = \Lambda_n v(x) e^{-\Lambda_n \bar{\nu}(x)}.$$

Integrating over $D$ and using the boundary limits gives

$$\Lambda_n \int_D v(x) e^{-\Lambda_n \bar{\nu}(x)}\, dx = e^{-\Lambda_n \bar{\nu}(r^-)} - e^{-\Lambda_n \bar{\nu}(l^+)} = 1 - 0 = 1.$$

Hence $\int_D v(x) e^{-\Lambda_n \bar{\nu}(x)}\, dx = 1/\Lambda_n$, and thus $\mathbb{P}(I_n = i \mid \boldsymbol{s}_n) = w_{i,n}/\Lambda_n\,(1 + o(1))$ uniformly in $i$. $\qquad \square$

### B.1. Computing tail masses from tail ratios under geometry-matched indexing

The previous proposition provides *tail ratios* in the extreme-value window. To apply Theorem 2.4 we also need a common-factor representation for the *tail masses* $w_{i,n} = \bar{F}_{s_{i,n}}(b_n)$ across the index $i$. The following lemma records the corresponding calculations under the three canonical geometry-matched index models (Table 1 in the main text).

**Theorem B.2** (Tail masses under geometry-matched index models). *Let $\{s_{i,n}\}_{i \in \mathcal{C}_n}$ be deterministic scores and let $w_{i,n} := \bar{F}_{s_{i,n}}(b_n)$.*

(a) **Translation (Gumbel-type indexing).** *Assume $F_s(x) = F_0(x - s)$ for a continuous baseline $F_0$. Assume there exist $b_n \in \mathbb{R}$, $a_n > 0$ and a function $\bar{\nu}$ such that for every compact $K \subset D$,*

$$\sup_{x \in K} \left| \frac{\bar{F}_0(b_n + a_n x)}{\bar{F}_0(b_n)} - \bar{\nu}(x) \right| \to 0.$$

*If $t_{i,n} := s_{i,n}/a_n$ satisfy $\sup_{i \in \mathcal{C}_n} |t_{i,n}| \leq T$ for some $T < \infty$ with $[-T, T] \subset D$, then uniformly in $i \in \mathcal{C}_n$,*

$$w_{i,n} = \bar{F}_0(b_n) \, \bar{\nu}(-t_{i,n}) \, (1 + o(1)).$$

*In particular, if $\bar{\nu}(x) = e^{-x}$ (Gumbel), then $w_{i,n} = \bar{F}_0(b_n) \exp(s_{i,n}/a_n)(1 + o(1))$.*

(b) **Scaling (Fréchet-type indexing).** *Assume $X_s = \rho(s) Z$ where $Z$ is a nonnegative baseline random variable with tail $\bar{F}_Z$ and $\rho(s) > 0$. Assume there exist $b_n \to \infty$ and $\alpha > 0$ such that for every compact $K \subset (0, \infty)$,*

$$\sup_{y \in K} \left| \frac{\bar{F}_Z(b_n y)}{\bar{F}_Z(b_n)} - y^{-\alpha} \right| \to 0.$$

*If $\rho(s_{i,n}) \in [\rho_{\min}, \rho_{\max}]$ for some $0 < \rho_{\min} \leq \rho_{\max} < \infty$, then uniformly in $i \in \mathcal{C}_n$,*

$$w_{i,n} = \bar{F}_Z(b_n) \, \rho(s_{i,n})^\alpha \, (1 + o(1)).$$

(c) **Endpoint shortfall (Weibull-type indexing).** *Assume there is a common right endpoint $b \in \mathbb{R}$ and*

$$b - X_s = q(s) \, Z,$$

*where $q(s) > 0$ and $Z \geq 0$ is a baseline random variable with CDF $H$. Assume there exist $\beta > 0$, $\lambda > 0$ and $z_0 > 0$ such that*

$$\sup_{0 < z \leq z_0} \left| \frac{H(z)}{\lambda z^\beta} - 1 \right| \to 0.$$

*Let $t_n \downarrow 0$ be any sequence such that $t_n/q(s_{i,n}) \leq z_0$ for all large $n$ and all $i \in \mathcal{C}_n$, and set $b_n := b - t_n$. Then uniformly in $i \in \mathcal{C}_n$,*

$$w_{i,n} = \mathbb{P}(X_{s_{i,n}} > b - t_n) = \lambda t_n^\beta \, q(s_{i,n})^{-\beta} \, (1 + o(1)).$$

*Proof.* **(a)** By the translation model,

$$w_{i,n} = \bar{F}_{s_{i,n}}(b_n) = \bar{F}_0(b_n - s_{i,n}) = \bar{F}_0(b_n + a_n(-t_{i,n})).$$

Since $-t_{i,n} \in [-T, T] \subset D$ uniformly, the assumed tail ratio gives

$$\frac{w_{i,n}}{\bar{F}_0(b_n)} = \frac{\bar{F}_0(b_n + a_n(-t_{i,n}))}{\bar{F}_0(b_n)} = \bar{\nu}(-t_{i,n}) + o(1)$$

uniformly in $i$.

**(b)** By $X_s = \rho(s)Z$,

$$w_{i,n} = \mathbb{P}(\rho(s_{i,n})Z > b_n) = \bar{F}_Z\left( \frac{b_n}{\rho(s_{i,n})} \right).$$

Set $y_{i,n} := 1/\rho(s_{i,n}) \in [1/\rho_{\max}, 1/\rho_{\min}]$, a compact subset of $(0, \infty)$. Then

$$\frac{w_{i,n}}{\bar{F}_Z(b_n)} = \frac{\bar{F}_Z(b_n y_{i,n})}{\bar{F}_Z(b_n)} = y_{i,n}^{-\alpha} + o(1) = \rho(s_{i,n})^{\alpha} + o(1),$$

uniformly in $i$.

**(c)** By $b - X_s = q(s)Z$,

$$
\begin{aligned}
w_{i,n} &= \mathbb{P}(X_{s_{i,n}} > b - t_n) = \mathbb{P}(b - X_{s_{i,n}} < t_n) \\
&= \mathbb{P}\left(Z < \frac{t_n}{q(s_{i,n})}\right) = H\left(\frac{t_n}{q(s_{i,n})}\right)
\end{aligned}
$$

Since $t_n/q(s_{i,n}) \leq z_0$ uniformly for all large $n$, the uniform expansion of $H$ yields

$$w_{i,n} = \lambda \left(\frac{t_n}{q(s_{i,n})}\right)^{\beta}(1 + o(1)) = \lambda t_n^{\beta} q(s_{i,n})^{-\beta}(1 + o(1)),$$

uniformly in $i$. $\qquad\square$

## C. Geometry mismatch and misspecification for top-1 links

### C.1. Quasi maximum likelihood under link misspecification

In the main text we compare different parametric top 1 links $p_\theta(\cdot \mid s)$ for winner observations. It is helpful to separate two cases.

**Correct specification.** If the data are generated from the assumed link family, meaning there exists a parameter $\theta_\star$ such that $p_\star(\cdot \mid s) = p_{\theta_\star}(\cdot \mid s)$ almost surely, then maximum likelihood targets $\theta_\star$ and, under standard regularity conditions, the estimator is asymptotically efficient.

**Misspecification and quasi maximum likelihood.** If the true conditional winner law $p_\star(\cdot \mid s)$ is not contained in the assumed family, then the maximum likelihood estimator is still well defined, but it no longer targets a true parameter. Instead, it converges to a *pseudo true* parameter

$$\theta^\star \in \arg\max_\theta \mathbb{E}\left[\log p_\theta(Y \mid s)\right],$$

which is the best approximation to $p_\star$ within the chosen link family (equivalently, it minimizes a KL divergence in expectation). In this case the estimator is called a quasi maximum likelihood estimator, and its large sample covariance is given by a robust sandwich expression. We refer to (White, 1982; Gourieroux et al., 1984; van der Vaart, 1998) for general statements and conditions. The formal statement is Theorem C.1; we include the proof sketch here for completeness.

**Theorem C.1** (Quasi maximum likelihood under misspecification, after White). *Let $Z_j = (Y_j, s_j)$ be i.i.d. from $P_\star$, and let $p_\theta(\cdot \mid s)$ be a possibly misspecified conditional top-1 link. If $\widehat{\theta}_m \in \arg\max_{\theta \in \Theta} \sum_{j=1}^m \log p_\theta(Y_j \mid s_j)$, then under standard regularity conditions $\widehat{\theta}_m \to \theta^\star$, where*

$$\theta^\star \in \arg\max_{\theta \in \Theta} \mathbb{E}_{P_\star} \log p_\theta(Y \mid s),$$

*and*

$$\sqrt{m}(\widehat{\theta}_m - \theta^\star) \Rightarrow \mathcal{N}(0, H^{-1}JH^{-1}).$$

*Here, with $\ell_\theta(Z) = \log p_\theta(Y \mid s)$, $H = -\mathbb{E}\nabla_\theta^2 \ell_{\theta^\star}(Z)$ and $J = \mathbb{E}[\nabla_\theta \ell_{\theta^\star}(Z)\nabla_\theta \ell_{\theta^\star}(Z)^\top]$. Correct specification gives $J = H$; under misspecification the estimator targets the pseudo-true KL projection.*

*Proof sketch of Theorem C.1.* This is a standard result for $M$ estimators. With $\ell_\theta(Z) = \log p_\theta(Y \mid s)$, the score equation $\sum_{j=1}^m \nabla_\theta \ell_{\widehat{\theta}_m}(Z_j) = 0$ is expanded around $\theta^\star$ via a Taylor approximation. A law of large numbers controls the Hessian term and a multivariate central limit theorem controls the score term, yielding the stated asymptotic normality with sandwich covariance. See (White, 1982; van der Vaart, 1998; Gourieroux et al., 1984). $\qquad\square$

**Finite-sample illustration.** The theorem says that a misspecified likelihood can become statistically stable around the wrong object: the pseudo-true KL projection. Figure 5 illustrates this phenomenon in the same top-1 language used in the paper. Data are generated from an endpoint-shortfall law, and the target is the true top-vs-reference log-odds. The correctly specified Weibit estimator has the expected $\mathcal{O}(1/n)$ error decay, while the misspecified one-parameter softmax estimator converges to its pseudo-true limit and retains a nonzero asymptotic error floor. Thus Figure 5 should be read as a concrete demonstration of Theorem C.1, specialized to the link misspecification that arises when endpoint geometry is forced into a softmax coordinate.

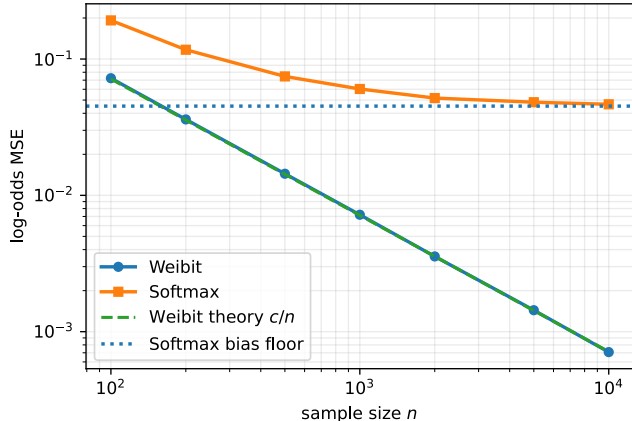

*Figure 5.* Noise floor from link misspecification in a synthetic top-1 model. Data are generated from an endpoint-shortfall (Weibit) law and the target is the true top-vs-reference log-odds. The correctly specified Weibit estimator has error decaying as $\Theta(1/n)$, while the misspecified one-parameter softmax converges to a pseudo-true parameter and retains a nonzero asymptotic error floor. This illustrates the practical consequence of Theorem C.1.

**From estimation to training.** The noise-floor example is an estimation-level consequence: the wrong link converges, but to the wrong target. In contrastive learning, the same top-1 link also determines how each batch allocates optimization pressure. The next subsection translates the same misspecification into a gradient-level diagnostic.

## C.2. Gradient misallocation under link misspecification

In our setting, $p_\theta(Y \mid s)$ is the top-1 link used by the contrastive loss. Thus a systematic held-out gain for the endpoint-blended link means that the softmax fit should be viewed as a pseudo-true projection rather than the data-generating winner law. For training, this projection has an operational consequence: it determines where gradient mass is placed. The InfoNCE gradient with respect to negative $j$ is $\partial\mathcal{L}/\partial s_j = p_j^{\mathrm{SM}}/\tau$, where $1/\tau$ is constant across negatives. The normalized softmax gradient is therefore exactly the model's predicted win-probability distribution $\{p_j^{\mathrm{SM}}\}$. For anchors whose top-1 outcomes are well described by the softmax/Gumbel link, this distribution should align with the empirical failure distribution. Systematic divergence is evidence that a pure softmax link is misspecified for a nontrivial part of the batch.

To measure this, we freeze an encoder and, for each anchor, repeatedly sample random subsets of $K$ negatives from a large bank of $N \approx 1536$. Whenever a negative beats the positive (a "failure"), we record the cosine similarity of the culprit, giving the **risk distribution** (black curve in Figure 6). On the same subsets, we compute the gradient magnitude of InfoNCE and of a pure endpoint-shortfall (Weibit) loss with respect to every negative and bin by score, giving the **gradient distributions** (red = softmax, blue = Weibit). The blue curve is a diagnostic reference for the endpoint regime; the training objective in Section 3 decides anchor by anchor how much of this component to use.

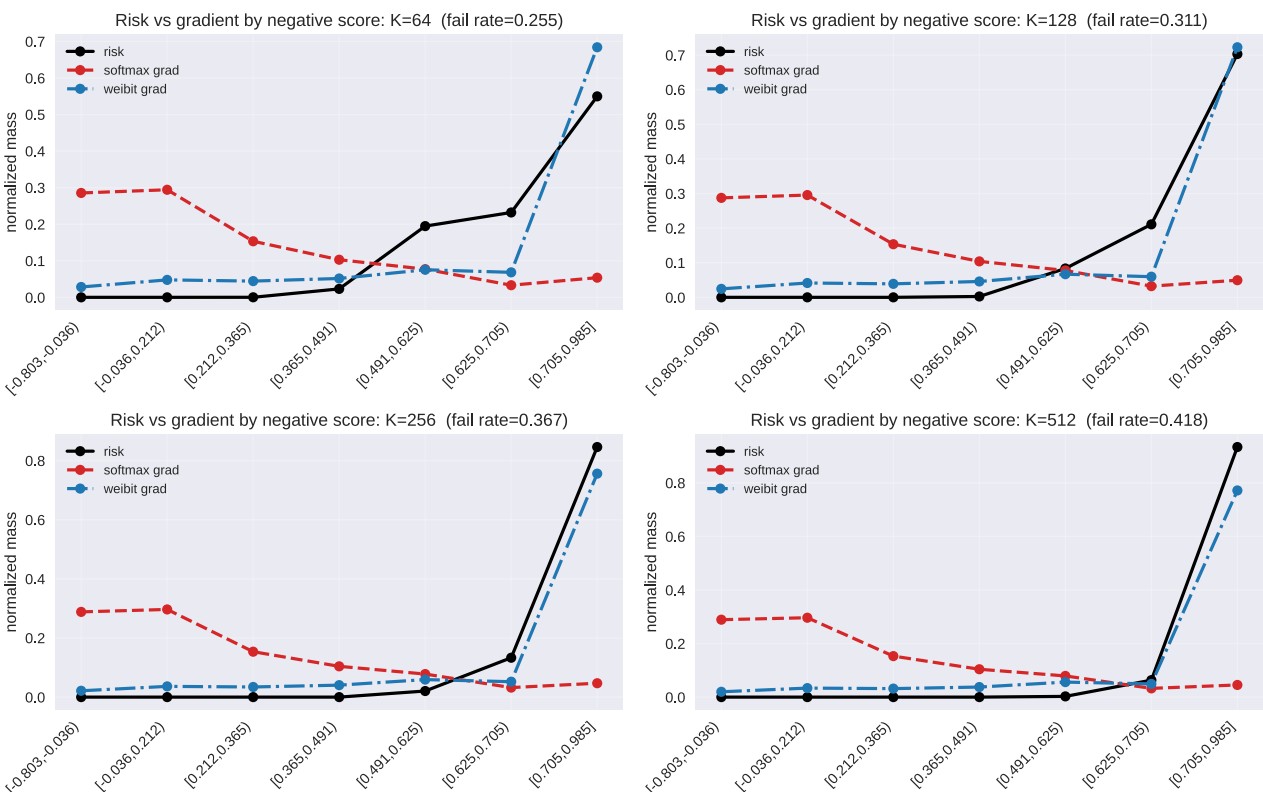

*Figure 6.* Risk vs. gradient distributions, binned by negative cosine similarity. Black: where failures come from. Red: where InfoNCE allocates gradient. Blue: where a pure endpoint-shortfall (Weibit) diagnostic allocates gradient. Each panel is a different choice-set size $K$.

Figure 6 reveals two facts. First, InfoNCE misallocates gradient mass: the risk (black) concentrates in the highest score bins, while the softmax gradient (red) spreads substantial mass across low and medium score bins where the empirical failure risk is nearly zero. This happens because the exponential function assigns non-trivial probability to every negative; with hundreds of easy negatives each getting a small share, their combined gradient mass can drown out the few genuinely dangerous hard ones. Second, the pure Weibit diagnostic (blue) is much closer to the empirical risk because it concentrates gradient mass near the finite endpoint. It is not identical to the black curve, however: depending on $K$, the empirical risk may retain mass in the second-highest bin or be even more concentrated in the final bin than the Weibit gradient. This residual mismatch is expected from the theory: FTG describes the extreme-tail domain of attraction, whereas a finite minibatch contains a mixture of anchors, some with near-cap competitors and some whose effective top-1 behavior is still closer to the translation coordinate. Thus the figure is evidence for a substantial endpoint component, not evidence that pure Weibit should replace softmax for every anchor. This motivates the corrected objective in Section 3, which preserves the softmax link when endpoint evidence is weak and redirects gradient mass toward the near-cap regime when the batch tail is Weibull-like.

### C.3. Coordinate mismatch implies link misspecification

We next record the elementary coordinate-mismatch facts that justify the link-misspecification interpretation used above. Changing the score coordinate inside a softmax generally changes the induced top 1 link in a way that cannot be fixed by re tuning the temperature. Thus, when the extreme tail is naturally described by an endpoint-shortfall coordinate, forcing it into the translation coordinate creates a genuine top-1 link mismatch.

We use the following shorthand for a softmax top-1 link in a score coordinate $T$:

$$p_i^{(T,\tau)}(\boldsymbol{s}) \triangleq \frac{\exp(T(s_i)/\tau)}{\sum_{j=0}^{K} \exp(T(s_j)/\tau)}, \qquad i \in \{0, \dots, K\},$$

where $s = (s_0, \ldots, s_K)$ and $\tau > 0$.

**Example 1** (Translation vs endpoint coordinate cannot be absorbed by temperature). Let $T_{\mathrm{tr}}(s) = s$ (translation coordinate) and $T_{\mathrm{ep}}(s) = -\log(1 - s)$ for $s < 1$ (endpoint shortfall coordinate). Consider three candidates with scores $(s_0, s_1, s_2) = (0, 0.5, 0.9)$. For any coordinate $T$ and any temperature $\tau$, the ratio of pairwise log-odds

$$R_T(s) \triangleq \frac{\log\left(p_1^{(T,\tau)}(s)/p_0^{(T,\tau)}(s)\right)}{\log\left(p_2^{(T,\tau)}(s)/p_0^{(T,\tau)}(s)\right)} = \frac{T(s_1) - T(s_0)}{T(s_2) - T(s_0)}$$

is independent of $\tau$. A direct computation gives

$$R_{T_{\mathrm{tr}}}(s) = \frac{0.5 - 0}{0.9 - 0} = \frac{5}{9},$$

$$R_{T_{\mathrm{ep}}}(s) = \frac{-\log(1 - 0.5)}{-\log(1 - 0.9)} = \frac{\log 2}{\log 10}$$

Since $\frac{5}{9} \neq \frac{\log 2}{\log 10}$, there is no choice of temperatures $\tau_{\mathrm{tr}}, \tau_{\mathrm{ep}} > 0$ such that $p^{(T_{\mathrm{tr}},\tau_{\mathrm{tr}})}(s) = p^{(T_{\mathrm{ep}},\tau_{\mathrm{ep}})}(s)$. In particular, a softmax link in the translation coordinate cannot reproduce a softmax link in the endpoint coordinate on score triples of this form.

**Lemma C.2** (Softmax coordinate equivalence forces an affine transform). *Let $I \subset \mathbb{R}$ and let $T, \widetilde{T} : I \to \mathbb{R}$ be functions. Assume there exist temperatures $\tau, \widetilde{\tau} > 0$ such that for every $K \geq 1$ and every score vector $s \in I^{K+1}$,*

$$p^{(T,\tau)}(s) = p^{(\widetilde{T},\widetilde{\tau})}(s).$$

*Then there exist constants $a > 0$ and $b \in \mathbb{R}$ such that*

$$T(s) = a\,\widetilde{T}(s) + b, \qquad \forall s \in I,$$

*with $a = \tau/\widetilde{\tau}$.*

*Proof.* It suffices to consider $K = 1$ (two candidates). Equality of the two links implies that for all $s_0, s_1 \in I$,

$$\frac{p_0^{(T,\tau)}(s_0, s_1)}{p_1^{(T,\tau)}(s_0, s_1)} = \frac{p_0^{(\widetilde{T},\widetilde{\tau})}(s_0, s_1)}{p_1^{(\widetilde{T},\widetilde{\tau})}(s_0, s_1)}.$$

By the softmax form,

$$\log\left(\frac{p_0^{(T,\tau)}(s_0, s_1)}{p_1^{(T,\tau)}(s_0, s_1)}\right) = \frac{T(s_0) - T(s_1)}{\tau}, \tag{C.1}$$

$$\log\left(\frac{p_0^{(\widetilde{T},\widetilde{\tau})}(s_0, s_1)}{p_1^{(\widetilde{T},\widetilde{\tau})}(s_0, s_1)}\right) = \frac{\widetilde{T}(s_0) - \widetilde{T}(s_1)}{\widetilde{\tau}} \tag{C.2}$$

Therefore, for all $s_0, s_1 \in I$,

$$T(s_0) - T(s_1) = \frac{\tau}{\widetilde{\tau}}\left(\widetilde{T}(s_0) - \widetilde{T}(s_1)\right).$$

Fix any $s_{\mathrm{ref}} \in I$ and set $s_1 = s_{\mathrm{ref}}$. Writing $a \triangleq \tau/\widetilde{\tau}$ and $b \triangleq T(s_{\mathrm{ref}}) - a\,\widetilde{T}(s_{\mathrm{ref}})$ yields

$$T(s) = a\,\widetilde{T}(s) + b, \qquad \forall s \in I.$$

$\square$

**Corollary C.3** (Mismatch in tail coordinate implies misspecification). *Let $\widetilde{T}(s) = s$ and let $T$ be any non-affine transform on $I$. Then there is no temperature $\tau > 0$ such that the link $p^{(T,\widetilde{\tau})}$ can be represented as $p^{(\widetilde{T},\tau)}$ uniformly over all candidate sets and score vectors.*

*Table 4.* Weibull-vs-softmax (nested) link selection across datasets/backbones. All values represent the mean $\pm$ 95% confidence interval half-width across seeds. $\hat{\lambda}$ is the fitted shortfall-mixture weight in the blended tail-coordinate link. $\overline{\Delta}_{W-0}$ is the held-out mean log-likelihood gain (nats/anchor) of the Weibull (shortfall) link over softmax. $e^{\overline{\Delta}_{W-0}}$ is the corresponding *geometric* improvement factor in the probability assigned to the observed top-1 (paired positive).

| Dataset | Backbone | $\hat{\lambda}$ | $\overline{\Delta}_{W-0}$ | $e^{\overline{\Delta}_{W-0}}$ | $n_{\text{seeds}}$ |
|---------|----------|-----------------|---------------------------|-------------------------------|--------------------|
| CIFAR-10 | ResNet-18 | $0.393 \pm 0.018$ | $0.352 \pm 0.006$ | $1.422 \pm 0.009$ | 7 |
| CIFAR-10 | ResNet-50 | $0.355 \pm 0.011$ | $0.360 \pm 0.010$ | $1.433 \pm 0.015$ | 10 |
| CIFAR-100 | ResNet-18 | $0.495 \pm 0.011$ | $0.223 \pm 0.005$ | $1.250 \pm 0.007$ | 10 |
| CIFAR-100 | ResNet-50 | $0.480 \pm 0.018$ | $0.228 \pm 0.010$ | $1.256 \pm 0.013$ | 10 |
| STL-10 | ResNet-18 | $0.330 \pm 0.035$ | $0.252 \pm 0.016$ | $1.286 \pm 0.021$ | 10 |
| STL-10 | ResNet-50 | $0.325 \pm 0.030$ | $0.270 \pm 0.018$ | $1.309 \pm 0.023$ | 10 |

## D. Likelihood Ratio Test Details

This appendix describes the likelihood-based link selection procedure used in Section 2.3. Table 4 shows the actual numbers in Figure 4. As a complementary analysis to Figure 4, we display the $\hat{\lambda}$ and $\Delta$ across epochs in Figure 9.

### D.1. Hard-negative and pool-size ablations

The main text reports the aggregate link-selection result. Here we decouple two effects: the hardness of the negatives included in the choice set and the size of the pool from which those hard negatives are drawn. When the total pool is fixed at $K = 512$, adding progressively easier negatives reduces both the fitted endpoint weight and the held-out likelihood gain (Figure 7). When the hard set is fixed at $K_{\text{neg}} = 64$ and the pool size grows, the selected hard negatives become closer to the endpoint and the fitted endpoint weight increases (Figure 8).

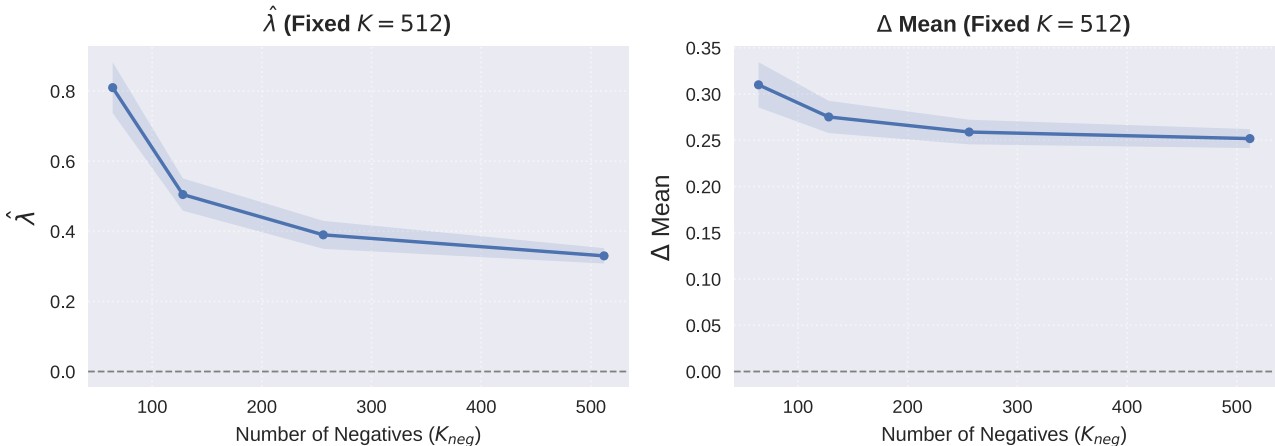

*Figure 7.* **Decoupling the choice set size from the negative pool (fixed $K = 512$).** For each anchor, we form a negative pool of size $K=512$, then construct the choice set by hard selecting $K_{\text{neg}}$ negatives (largest similarity) and adding the positive. Left: fitted $\hat{\lambda}$. Right: held-out $\overline{\Delta}$. As $K_{\text{neg}}$ increases, the set includes progressively less extreme negatives, so both $\hat{\lambda}$ and $\overline{\Delta}$ decrease.

### D.2. Data, candidate sets, and top-1 outcomes

For a frozen checkpoint, we construct an evaluation bank of $N_{\text{batch}} = 32$ batches of size $B = 256$. Each batch contains $2B$ augmented views. For each anchor view $i \in \{1, \dots, 2B\}$, let $i^+$ denote the paired positive view. The candidate set is

$$\mathcal{C}(i) = \{1, \dots, 2B\} \setminus \{i\},$$

and we treat the observed top-1 outcome as $Y_i = i^+$. We compute cosine similarity scores $s_{ik}$ between the normalized embeddings of $i$ and $k$ for all $k \in \mathcal{C}(i)$. This yields a conditional choice dataset $\{(\{s_{ik}\}_{k \in \mathcal{C}(i)}, Y_i = i^+)\}$.

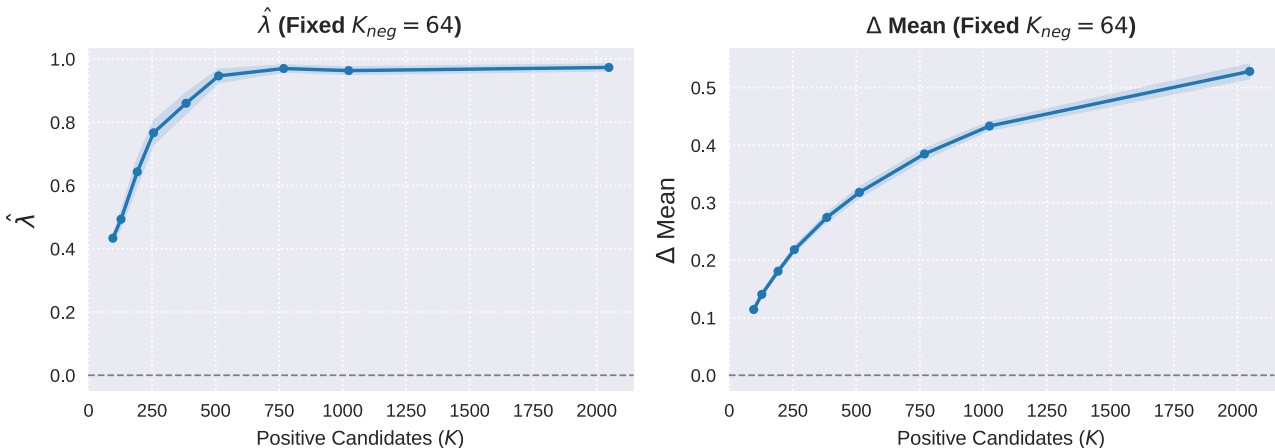

*Figure 8.* **Scaling the negative pool with a fixed hard set** ($K_{\mathrm{neg}} = 64$)**.** We fix the number of hard negatives at $K_{\mathrm{neg}}{=}64$ and vary the pool size $K$. Left: fitted $\hat{\lambda}$. Right: held-out $\overline{\Delta}$. As $K$ grows, the hardest negatives become more extreme and $\hat{\lambda}$ increases monotonically toward one.

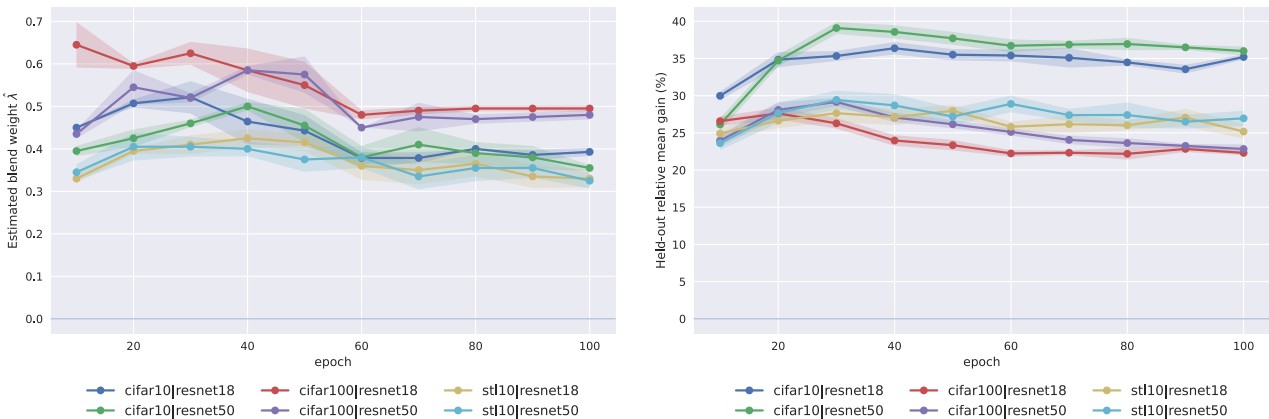

*Figure 9.* Likelihood based link selection across epochs, datasets and backbones complementary to Figure 4. Top: estimated blend weight $\hat{\lambda}$ (mean across seeds, 95% confidence interval) across epochs. Bottom: held out mean log likelihood gain $\overline{\Delta}$ (nats per anchor; mean across seeds, 95% confidence interval).

### D.3. Models and parameter fitting

We compare the null softmax link (2.7) to the nested blended-coordinate link (2.9)–(2.8). We split batches into held-in and held-out sets (using `train_frac = 0.5`). Parameters are fit by maximizing the held-in top-1 log-likelihood:

$$\hat{\tau}_0 = \arg\max_{\tau_0 > 0} \sum_{i \in \text{held-in}} \log p_0(Y_i = i^+ \mid \{s_{ik}\}_{k \in \mathcal{C}(i)})$$

and

$$(\hat{\lambda}, \hat{\tau}_1) = \arg\max_{\substack{\lambda \in [0,1] \\ \tau_1 > 0}} \sum_{i \in \text{held-in}} \log p_1(Y_i = i^+ \mid \{s_{ik}\}_{k \in \mathcal{C}(i)})$$

In our runs we use a grid search with `lam_grid_size = 21` and `tau_grid_size = 21`.

## D.4. Held-out effect size

On held-out batches we compute the per-batch mean log-likelihood difference

$$\Delta_b \triangleq \frac{1}{2B} \sum_{i=1}^{2B} \Big( \log p_1(Y_i = i^+) - \log p_0(Y_i = i^+) \Big), \tag{D.1}$$

and report the held-out mean

$$\overline{\Delta} \triangleq \frac{1}{|\mathcal{B}_{\text{test}}|} \sum_{b \in \mathcal{B}_{\text{test}}} \Delta_b.$$

Because $\overline{\Delta}$ is a mean log-ratio, $\exp(\overline{\Delta})$ equals the ratio of geometric-mean probabilities assigned to the positive under the two links.

## D.5. Significance tests and uncertainty quantification

**Clustered (batch-level) $t$-test.** Anchors within the same batch share the same negative pool, hence their log-likelihood terms are dependent. We therefore treat each batch as an approximately independent *cluster* and test

$$H_0 : \mathbb{E}[\Delta_b] \leq 0 \qquad \text{vs.} \qquad H_1 : \mathbb{E}[\Delta_b] > 0$$

using a one-sided $t$-test over the batch-level statistics $\{\Delta_b\}$.

**Nonparametric bootstrap over batches (95% CI).** To obtain confidence intervals for $\overline{\Delta}$ without relying on normality, we resample held-out batches with replacement and recompute $\overline{\Delta}$ on each bootstrap replicate. The 2.5% and 97.5% percentiles give a nonparametric 95% CI. (Resampling at the batch level preserves the within-batch dependence structure.)

**Parametric bootstrap for a nested likelihood ratio test.** Since $\mathcal{H}_0$ is nested in $\mathcal{H}_1$ at the boundary $\lambda = 0$, standard asymptotic $\chi^2$ calibration for the likelihood ratio statistic may be inaccurate in finite samples. We therefore optionally calibrate the likelihood ratio test using a parametric bootstrap: (i) fit $\hat{\tau}_0$ on held-in data under $\mathcal{H}_0$, (ii) simulate synthetic winners $Y_i$ from $p_0(\cdot \mid \hat{\tau}_0)$ while keeping score vectors $\{s_{ik}\}$ fixed, (iii) refit both $\mathcal{H}_0$ and $\mathcal{H}_1$ on the simulated held-in data, compute the simulated LRT statistic, and (iv) repeat to form an empirical null distribution and a bootstrap $p$-value.

**$K$-fold cross-validation.** To reduce sensitivity to a single train/test split, we also run `cv_k = 5` fold cross-validation over batches. For each fold, we fit parameters on the training folds and evaluate $\overline{\Delta}$ on the held-out fold. We report the mean and standard deviation across folds.

## D.6. Fréchet ablation

**What is being tested.** For each anchor $i$ we observe a candidate set $\mathcal{N}_i$ and a *score vector* $\{s_{ij}\}_{j \in \mathcal{N}_i}$ produced by a frozen encoder checkpoint, along with a top-1 outcome (the paired positive $i^+$). We *do not* treat these scores as latent utilities. Instead, we treat the score vector as the conditioning statistic and compare *link functions* $p(y = i^+ \mid \{s_{ij}\})$ by held-out top-1 log-likelihood. Since cosine similarities satisfy $s_{ij} \in [-1, 1]$, a bounded-endpoint (shortfall) geometry is a natural candidate for the observed statistic, whereas a heavy-tail (Fréchet) geometry is a priori less plausible; we nevertheless include it as an explicit ablation below.

**Test statistic.** Given two candidate links $p_A$ and $p_B$, we report the held-out mean log-likelihood gain

$$\overline{\Delta}_{A-B} \triangleq \frac{1}{|\mathcal{D}_{\text{test}}|} \sum_{(i, \mathcal{N}_i) \in \mathcal{D}_{\text{test}}} \Big[ \log p_A(i^+ \mid \{s_{ij}\}_{j \in \mathcal{N}_i}) - \log p_B(i^+ \mid \{s_{ij}\}_{j \in \mathcal{N}_i}) \Big]$$

measured in nats/anchor. We aggregate over multiple random seeds by averaging $\overline{\Delta}$ within each seed and reporting the mean $\pm$ 95% confidence intervals across seeds.

**Conclusion.** Across datasets/backbones, including an endpoint shortfall coordinate yields a robust increase in held-out top-1 likelihood (Table 4, Figure 10 top). By contrast, a heavy-tail Fréchet component provides no measurable improvement either by itself (middle) or when added on top of the Weibull model (bottom). This supports our focus on a Gumbel + Weibull blend (softmax + shortfall) rather than a three-way mixture including Fréchet.

*Table 5.* Fréchet ablation summary. All values represent the mean $\pm$ 95% confidence interval half-width. $\overline{\Delta}_{F-0}$ is the held-out mean gain of a *Fréchet-only* (heavy-tail) link over softmax. $\overline{\Delta}_{WF-W}$ is the incremental gain from adding a Fréchet component on top of the Weibull (shortfall) model. $\hat{\lambda}_F^{(\text{full})}$ is the fitted Fréchet mixture weight in the joint (Weibull+Fréchet) model. The empirical finding is that Fréchet provides no measurable benefit once shortfall geometry is included ($\overline{\Delta}_{WF-W} \approx 0$).

| Dataset | Backbone | $\hat{\lambda}_W$ | $\overline{\Delta}_{W-0}$ | $\overline{\Delta}_{F-0}$ | $\overline{\Delta}_{WF-W}$ | $\hat{\lambda}_F^{(\text{full})}$ |
|---------|----------|-------------------|---------------------------|---------------------------|----------------------------|-----------------------------------|
| CIFAR-100 | ResNet-18 | $0.490 \pm 0.015$ | $0.206 \pm 0.002$ | $0.000 \pm 0.000$ | $0.000 \pm 0.002$ | $0.075 \pm 0.039$ |
| STL-10 | ResNet-18 | $0.350 \pm 0.038$ | $0.253 \pm 0.017$ | $0.000 \pm 0.000$ | $0.000 \pm 0.001$ | $0.305 \pm 0.148$ |

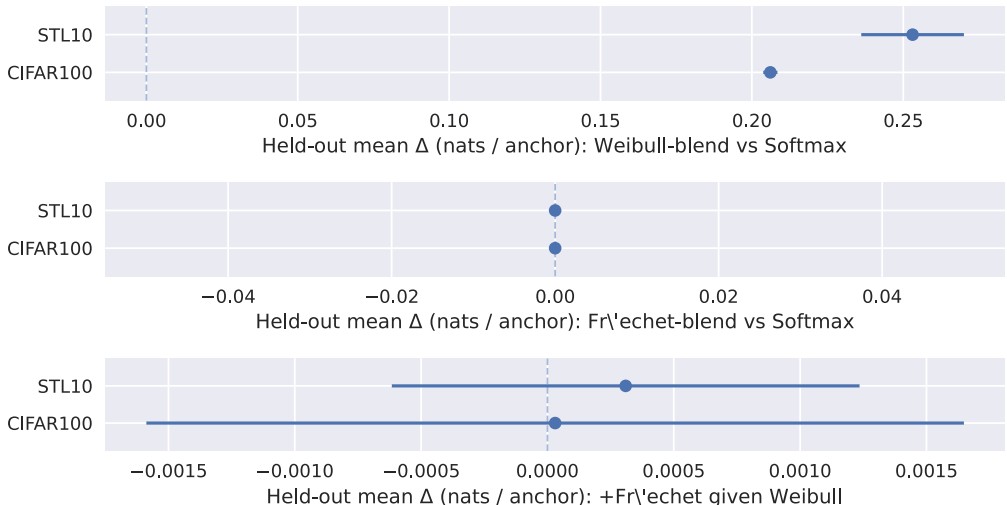

*Figure 10.* Forest plots of held-out log-likelihood gains (mean $\pm$ 95% CI over seeds). **(top)** $\overline{\Delta}_{W-0}$: Weibull (shortfall) link vs. softmax. **(middle)** $\overline{\Delta}_{F-0}$: Fréchet-only link vs. softmax. **(bottom)** $\overline{\Delta}_{WF-W}$: additional gain from adding Fréchet on top of Weibull. Across datasets/backbones, Weibull yields consistent positive gains, while Fréchet-only and the incremental Fréchet gain given Weibull are near zero.

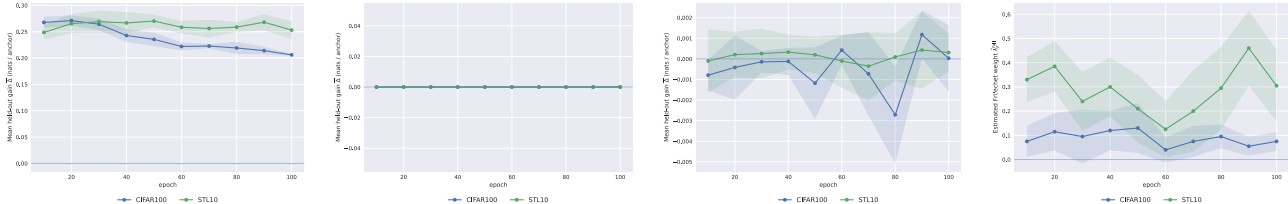

*Figure 11.* Per-epoch view of model selection quantities (mean $\pm$ 95% CI over seeds). From left to right: (1) Weibull-vs-softmax gain $\overline{\Delta}_{W-0}$ remains positive across epochs; (2) Fréchet-only gain $\overline{\Delta}_{F-0}$ stays near zero; (3) incremental gain $\overline{\Delta}_{WF-W}$ stays near zero, indicating that adding a Fréchet component does not improve predictive top-1 likelihood once shortfall geometry is included; (4) fitted Fréchet weight $\hat{\lambda}_F^{(\text{full})}$ in the joint model.

# E. Details of WEINCE

Algorithm 1 gives the main training procedure. Here we provide the derivation of the shortfall coordinate and additional implementation details.

**Deriving the endpoint shortfall coordinate.** In the Weibull indexing model in Table 1 we write

$$b - X_s = q(s)\, W, \qquad W \geq 0,$$

so that the asymptotic top 1 weights satisfy $p_i \propto q(s_i)^{-\beta}$ (Theorem 4.3 and Lemma B.3(c)). In our bounded similarity setting, the relevant endpoint is the score cap, so we take $b = x_F$ (for cosine similarity, $x_F = 1$). It is then natural to assume

that $q(s) > 0$ for $s < x_F$ and that $q(s) \to 0$ as $s \uparrow x_F$, meaning that the utility shortfall vanishes as the score approaches the cap.

Moreover, since the Weibull component of WEINCE targets the extreme regime $s \uparrow x_F$, only the local behavior of $q$ near $x_F$ is identifiable. If $q$ is left differentiable at $x_F$ with $c := -q'(x_F) > 0$, then a first order expansion gives

$$q(s) = c\,(x_F - s) + o(x_F - s) \qquad (s \uparrow x_F).$$

The multiplicative constant $c$ does not affect the induced choice probabilities, since it contributes a common factor $c^{-\beta}$ to $q(s)^{-\beta}$ (equivalently, an additive constant $-\beta \log c$ in the logits) that cancels after normalization. Therefore, without loss of generality we set

$$q(s) = x_F - s \quad (\text{cosine: } q(s) = 1 - s),$$

which yields the Weibull shortfall logits $\ell^W(s) = -\beta \log(x_F - s)$ used by WEINCE.

**Computational overhead.**    As reported in the main text, WEINCE adds no forward passes or trainable parameters; in the representative Tiny-ImageNet/ResNet-18 timing run, the additional tail fitting changes the end-to-end step time by less than $0.7\%$.

**Estimating $\lambda_i$ and the tail index $\beta_i$.**    We estimate $\lambda_i$ online from the current batch using two signals computed from the negative scores of anchor $i$. Let $\mathcal{N}(i)$ be the negatives and define shortfalls $\delta_{ij} = 1 - s_{ij}$.

*(i) Cap proximity (hardness).* Define the closest negative shortfall

$$\rho_i \triangleq \min_{j \in \mathcal{N}(i)} \delta_{ij} = 1 - \max_{j \in \mathcal{N}(i)} s_{ij}. \tag{E.1}$$

Small $\rho_i$ indicates that anchor $i$ is operating near the score ceiling.

*(ii) Tail shape evidence and $\beta_i$ (Weibull versus Gumbel proxy).* Let $\delta_{i,(1)} \leq \delta_{i,(2)} \leq \cdots$ be the ordered negative shortfalls, so that $\delta_{i,(1)} = \rho_i$. Using the $K_{\text{tail}}$ smallest shortfalls, we form an empirical CDF $\widehat{F}(\delta_{i,(k)}) \approx k/(K_{\text{tail}} + 1)$ and fit two one dimensional tail models:

$$\textbf{(Weibull):} \quad \log \widehat{F}(\delta_{i,(k)}) \approx a_i + \beta_i \log \delta_{i,(k)}, \tag{E.2}$$

$$\textbf{(Gumbel proxy):} \quad \log\big(-\log \widehat{F}(\delta_{i,(k)})\big) \approx c_i - \kappa_i \log \delta_{i,(k)}. \tag{E.3}$$

From the corresponding residual sums of squares we compute an AIC style score for each fit (up to additive constants) and define

$$\Delta\text{AIC}_i \triangleq \text{AIC}_G(i) - \text{AIC}_W(i), \tag{E.4}$$

so that $\Delta\text{AIC}_i > 0$ favors a Weibull style tail geometry for anchor $i$. We use the Weibull slope estimate $\hat{\beta}_i$ in the shortfall logits $\ell_{ij}^{\text{W}} = -\hat{\beta}_i \log(1 - s_{ij})$.

*(iii) Soft decision rule.* We convert these signals into a smooth interpolation weight via sigmoids:

$$\lambda_i \triangleq \sigma\big(\kappa_\rho(\log \rho_0 - \log \rho_i)\big)\,\sigma\big(\kappa_{\text{AIC}}(\Delta\text{AIC}_i - m)\big), \tag{E.5}$$

where $\sigma(\cdot)$ is the logistic sigmoid, $\rho_0$ is a pivot (for example, a batch quantile of $\{\rho_i\}$), $m \geq 0$ is an AIC margin, and $\kappa_\rho, \kappa_{\text{AIC}} > 0$ tune the transition sharpness.

**Practical details.**    We compute the shortfall term with clipping for numerical stability:

$$\log(1 - s_{ij}) \leftarrow \log\big(\varepsilon + 1 - \min(s_{ij},\, 1 - \varepsilon)\big)$$

for a small $\varepsilon > 0$. The statistics $\lambda_i$ and $\hat{\beta}_i$ are computed from the batch similarity matrix and treated as non differentiable per step estimates (that is, gradients are not backpropagated through $\lambda_i$ or $\hat{\beta}_i$).

*Table 6.* **kNN classifier accuracy at** $k = 50$ **(%).** We report the mean top-1 accuracy $\pm$ 95% confidence interval halfwidth for the baseline (INFONCE) and our method (WEINCE). Encoders are abbreviated as R18 (ResNet18), R50 (ResNet50), and ViT-S (ViT-Small). Best results are highlighted in bold.

| Dataset | Enc. | kNN Acc | |
|---|---|---|---|
| | | INFONCE | WEINCE |
| STL10 | R18 | $73.13 \pm 0.79$ | $\mathbf{74.68 \pm 0.59}$ |
| | R50 | $73.93 \pm 0.51$ | $\mathbf{75.72 \pm 0.54}$ |
| CIFAR10 | R18 | $78.20 \pm 0.39$ | $\mathbf{79.60 \pm 0.25}$ |
| | R50 | $77.98 \pm 0.36$ | $\mathbf{79.76 \pm 0.42}$ |
| CIFAR100 | R18 | $40.73 \pm 0.42$ | $\mathbf{47.38 \pm 0.28}$ |
| | R50 | $39.51 \pm 0.53$ | $\mathbf{46.41 \pm 0.24}$ |
| Tiny-IN | R18 | 24.41 | $\mathbf{25.13}$ |
| | ViT-S | 25.92 | $\mathbf{31.81}$ |

*Table 7.* **k-Nearest Neighbor (kNN) recall (R@50, %).** We report the mean recall $\pm$ 95% confidence interval halfwidth for the baseline (INFONCE) and our method (WEINCE). Encoders are abbreviated as R18 (ResNet18) and R50 (ResNet50). Best results are highlighted in bold.

| Dataset | Enc. | R@50 | |
|---|---|---|---|
| | | INFONCE | WEINCE |
| STL10 | R18 | $99.48 \pm 0.05$ | $\mathbf{99.54 \pm 0.08}$ |
| | R50 | $99.49 \pm 0.09$ | $\mathbf{99.50 \pm 0.14}$ |
| CIFAR10 | R18 | $99.06 \pm 0.21$ | $\mathbf{99.47 \pm 0.07}$ |
| | R50 | $99.06 \pm 0.05$ | $\mathbf{99.46 \pm 0.07}$ |
| CIFAR100 | R18 | $90.75 \pm 0.34$ | $\mathbf{92.81 \pm 0.31}$ |
| | R50 | $90.35 \pm 0.45$ | $\mathbf{92.43 \pm 0.39}$ |
| Tiny-IN | R18 | 76.46 | $\mathbf{76.70}$ |

**Drop in replacement for InfoNCE.** WEINCE keeps the SimCLR pipeline unchanged (augmentations, encoder, projection head, batch construction, and the same cross entropy over a $2N \times (2N - 1)$ similarity matrix). The only modification is the logit construction: after computing cosine similarities $s_{ij}$, we estimate, for each anchor, a tail index $\hat{\beta}_i$ and an interpolation weight $\lambda_i$ from batch tail statistics of the negative shortfalls $\delta_{ij} = 1 - s_{ij}$, then replace the vanilla logits $s_{ij}/\tau$ by the interpolated logits in (3.1). No additional learnable parameters are introduced beyond the SimCLR network, and all tail quantities are recomputed on the fly from the current minibatch similarity matrix and treated as stop gradient statistics.

# F. Details for Experiments

**Codebase.** All experiments use the official SimCLR PyTorch implementation from Spijkervet et al. (github.com/Spijkervet/SimCLR) as the training/evaluation pipeline. We keep the dataset-specific SimCLR settings from that codebase (augmentations, optimizer, temperature, schedule, etc.) fixed, and change *only* the contrastive loss (InfoNCE vs. WEINCE). The code will be released in github.com/hsme98/weince.

## F.1. Validation split and model selection

A key change relative to earlier drafts is that we use a held-out *validation* split carved out of the labeled training set, and we use this validation split for model selection. Concretely, for each dataset we construct a **stratified 80/20 split** of the labeled training set: 80% is used for training the downstream evaluator (linear probe) / building the kNN feature bank, and 20% is reserved as a validation set. The split is performed independently within each class by shuffling the indices of that class and assigning the first $\lfloor 0.2\, n_c \rfloor$ examples to validation and the remainder to training.

**What the validation set is used for.** When multiple hyperparameter configurations are considered (e.g., multiple WEINCE hyperparameter settings and/or multiple checkpoints), we select the configuration using **validation performance only**. The reported tables show the corresponding **test-set** performance of the configuration chosen on validation (no test-set tuning).

## F.2. Backbones and contrastive pretraining

We consider ResNet–18 and ResNet–50 backbones, each followed by the standard SimCLR projection head. For each dataset/backbone/loss configuration we train **four** independent encoders (four different pretraining seeds).

We use the default temperature $\tau = 0.5$ and optimizer settings (Adam with weight decay $10^{-6}$) provided in (Spijkervet, 2020), as suggested to be a good temperature choice across batch sizes. We trained encoders for 100 epochs. Following the default configuration of the SimCLR PyTorch codebase, we use a batch size of 256, and a ResNet backbone with $\ell_2$ normalization enabled and a projection head of output dimension 64. We use the standard SimCLR two-view augmentation pipeline and train with in-batch negatives (each batch contains $2N$ augmented views).

**Losses.** The baseline is the standard NT-Xent / InfoNCE objective. For WEINCE, we use the full logit-interpolation loss described in the main text (Equations (3.1) and (3.2) and Algorithm 1), where each anchor has an online-estimated interpolation weight $\lambda_i$ and tail index $\hat{\beta}_i$ computed from batch tail statistics. These per-step statistics are treated as stop-gradient quantities (no backpropagation through $\lambda_i$ or $\hat{\beta}_i$). Aside from the modified logits, the SimCLR pipeline is unchanged.

## F.3. Linear downstream classification (linear probe)

We evaluate the pretrained representation quality using a frozen-feature linear probe.

**Feature extraction.** Given a pretrained checkpoint, we freeze the encoder and compute feature vectors using the *encoder representation* $h$ (the backbone output before the projection head; in our code this is the first output returned by the SimCLR forward pass). Features are extracted using the repository's evaluation transform (`TransformsSimCLR(...).test_transform`) on: (i) the labeled training split and (ii) the held-out test split.

**Train/validation split (20%).** From the labeled training split we form a **stratified 80/20** train/validation partition (Section F.1). The linear probe is trained on the 80% partition and evaluated on both the 20% validation partition and the test set.

**Linear classifier and optimization.** We train a single linear classifier (multinomial logistic regression; a single fully-connected layer) on top of the frozen features with cross-entropy loss. We optimize the probe using Adam with learning rate $3 \times 10^{-4}$ for 100 epochs (batch size 256). No encoder weights are updated during this stage.

**Probe seeds and aggregation.** To account for probe-level randomness (initialization, minibatch order, optimizer noise), we train **five** probes per pretrained encoder using evaluation seeds $\{1, 2, 3, 4, 5\}$. This yields $4 \times 5 = 20$ linear-probe runs per dataset/backbone/loss. For each pretrained encoder seed $e$ we average the five probe runs to obtain a per-encoder mean accuracy $\mu_e$ (and the corresponding mean validation accuracy). We then treat the four encoder seeds as i.i.d. replicates and report the mean and 95% confidence intervals across encoder seeds (using a $t$-interval).

**Model selection.** When selecting among hyperparameter configurations (Section F.1), we use the validation accuracy (averaged over probe seeds for each encoder) to choose the configuration, and we report the corresponding test accuracy.

## F.4. kNN evaluation

We also evaluate the frozen representation using a non-parametric $k$-nearest-neighbor procedure.

**Feature extraction and similarity.** We use the same frozen encoder features $h$ as for linear probing, and $\ell_2$-normalize them. Nearest neighbors are computed using cosine similarity (dot product of normalized features).

**kNN bank, queries, and validation split.** The kNN *feature bank* is built from the 80% training partition of the labeled training set (Section F.1). We evaluate: (i) on the 20% validation partition (queries = validation, bank = training) and (ii) on the test set (queries = test, bank = training). This ensures that any selection based on validation does not leak test labels.

**Metrics.** We report **recall at** $k$ (R@$k$) of the extracted features for $k \in \{1, 2, 5, 10, 20, 50\}$, where

$$\text{R@}k = \frac{1}{|\mathcal{Q}|} \sum_{q \in \mathcal{Q}} \mathbf{1}\{\text{label}(q) \text{ appears in top-}k \text{ neighbors}\}$$

in Table 3. The main text reports R@1 through R@20 in Table 3; for runs where it was logged, the remaining R@50 column is displayed in Table 7. Our evaluation code additionally computes the standard SimCLR-style temperature-weighted kNN classification accuracy using $k = 50$ and temperature $\tau = 0.1$; this is not required to define R@$k$ but can be used as an auxiliary score. The results are provided in Table 6.

**Split seeds and aggregation.** For robustness, we repeat the stratified 80/20 train/validation split for kNN using five different split seeds $\{1, 2, 3, 4, 5\}$, and average the resulting validation/test kNN metrics across these splits. When multiple hyperparameter configurations are compared, selection is performed using the validation kNN metric (with the same averaging over split seeds), and the reported numbers are the test kNN metrics of the selected configuration.

