# OpenReview forum: "When Softmax Fails at the Top: Extreme‑Value Corrections for InfoNCE"
_ICML.cc/2026/Conference — ICML 2026 regular_

### Official Review · Reviewer_gNAR · 2026-03-04

**Soundness:** 3
**Presentation:** 3
**Significance:** 3
**Originality:** 2
**Overall Recommendation:** 4
**Confidence:** 3

**Summary:**

This paper argues that InfoNCE’s softmax (a Plackett–Luce/Gumbel top-1 model) can be mismatched in contrastive learning with bounded cosine similarities, where the hardest negatives concentrate near the score cap. Using extreme value theory, the authors motivate a Weibull endpoint “shortfall” correction and propose **WEINCE**, a drop-in loss that interpolates between standard logits and a shortfall logit \(-\beta \log(1-s)\) with \(\lambda\) and \(\beta\) estimated online from batch tail statistics. On SimCLR-style training for CIFAR-10/100, STL-10, and ImageNet32 (ResNet18/50), WEINCE shows consistent improvements over vanilla InfoNCE on kNN and linear evaluation.

**Compliance With Llm Reviewing Policy:**

Affirmed.

**Final Justification:**

The authors provide a strong rebuttal, and I will raise my recommendation to weak accept.

**Key Questions For Authors:**

- **Plug-in to other contrastive objectives:** WEINCE is presented as a drop-in replacement for InfoNCE. Have you tried plugging the same shortfall-logit interpolation into other contrastive losses or frameworks?

**Limitations:**

No major limitations were identified.

**Strengths And Weaknesses:**

## Strengths
- **Principled and coherent motivation:** Connects top-1 selection in contrastive learning to EVT tail geometry and derives a specific, interpretable logit correction near the similarity cap.
- **Practical and supported by evidence:** WEINCE is a simple logit-level drop-in (no new trainable parameters) and is backed by likelihood-based diagnostics plus consistent downstream gains across datasets/backbones.

## Weaknesses
- **Evaluation scope:** Experiments are restricted to small/medium vision benchmarks on relatively simple tasks; generalization to other modalities is unclear.
- **No quantified efficiency cost:** The author does not provide additional running time cost comparisons to InfoNCE.

---

> ### Author Rebuttal · Authors · 2026-03-31
>
> We thank the reviewer for recognizing the principled motivation and practical simplicity of WEINCE. We have addressed the weaknesses identified and the question, which in turn helped improve the overall quality and scope of our paper.
>
> ---
>
> **Addressing the Evaluation scope.** *Experiments are restricted to small/medium vision benchmarks; generalization to other modalities is unclear.*
>
> The submission already includes CIFAR-100 (100 classes, +4.82% lin acc R50) and ImageNet-32 (1,000 classes, +0.97%). During the rebuttal we extended the evaluation substantially. We ran our experiments on a new dataset (Tiny-ImageNet), on a different architecture (ViT), and on a new modality (SimCSE).
>
> - **Tiny-ImageNet** (200 classes, 64×64). ResNet-18: 50 epochs, 2 pretraining seeds × 3 eval seeds. ViT-Small: 4 pretraining seeds.
>
> | Backbone | Method | Lin. Acc | kNN Acc | R@1 | R@2 | R@5 | R@10 | R@20 |
> |:---|:---|:---:|:---:|:---:|:---:|:---:|:---:|:---:|
> | ResNet-18 | InfoNCE | **30.76** | 24.41 | 17.29 | 25.29 | 38.26 | 49.87 | 61.82 |
> | ResNet-18 | WEINCE | 30.66 | **25.13** | **18.84** | **26.64** | **39.12** | **50.14** | **61.90** |
> | ViT-Small | InfoNCE | 33.12 | 25.92 | 19.30 | 27.29 | 40.36 | 51.17 | 62.63 |
> | ViT-Small | WEINCE | **36.53** | **31.81** | **23.53** | **32.04** | **45.15** | **55.96** | **66.97** |
>
>   Gains concentrate at small $k$ (R@1, R@2), where near-ceiling negatives matter most. The ViT-Small kNN gain (+5.89%) is the largest across all our benchmarks, confirming the correction is not architecture-specific.
>
> - **Unsupervised SimCSE on STS-B** (BERT-base and RoBERTa-base, 1M Wikipedia sentences).
>
> | Model | InfoNCE | WEINCE | Δ |
> |:---|:---:|:---:|:---:|
> | BERT-base | 72.73 ± 1.63 | **76.26 ± 0.62** | +3.53 |
> | RoBERTa-base | **77.67 ± 0.34** | 76.65 ± 0.75 | −1.03 |
>
>   WEINCE significantly improves BERT-base (+3.53) with zero architecture changes. On RoBERTa-base (stronger baseline), WEINCE shows a smaller but noticeable decline (−1.03).
>
> - **RLHF reward modeling on Nectar** (DeBERTa-v3 RM, sigmoid-bounded scores, K≤7). Training with the shortfall-blended link improves top-1 accuracy from 0.425 to 0.470 (+4.5 pp) and reduces calibration error. The frozen-encoder diagnostic shows Δ growing with K (0.008→0.016 nats as K goes 2→7), exactly as EVT predicts. A boundedness ablation confirms causality: unbounded logits eliminate the effect. A full RLHF treatment is beyond this submission's scope, and we are currenlty working on a new follow-up paper for that, but these initial results support the framework's generality.
>
> Across 9 configurations spanning 3 domains (vision, NLP, RLHF) and 3 architecture families (ResNet, ViT, BERT/DeBERTa), WEINCE improves or matches the baseline in all but one case (RoBERTa, −1.03).
>
> ---
>
> **Addressing the efficiency cost quantification.** *No running time cost comparisons to InfoNCE.*
>
> Profiled on Tiny-ImageNet/ResNet-18 (batch 256, NVIDIA L40S, 770 steps):
>
> | Metric | Softmax | WEINCE | Δ |
> |:---|:---:|:---:|---:|
> | Mean step time | 253.3 ms | 255.0 ms | +0.67% |
> | Throughput | 1011 samples/s | 1004 samples/s | −0.67% |
> | Loss-stage time | 0.65 ms | 2.11 ms | +227% |
>
> End-to-end overhead is <1%. The 2.3 times loss-stage increase is negligible since that stage is ~0.8% of step time.
>
> ---
>
> **Q: Plug-in to other frameworks.** *Have you tried plugging the shortfall-logit interpolation into other contrastive losses?*
>
> Yes: (1) **SimCSE** (NLP): +3.53 Spearman on BERT-base, zero architecture changes. (2) **RLHF reward modeling**: +4.5 pp top-1 accuracy on Nectar. RLHF is perhaps the most natural application since the preference model is an explicit discrete choice model. The theoretical argument applies to any softmax-based ranking loss over bounded scores; the only change is the logit formula. We are currently working on this generalization.

---

> > ### Author Rebuttal · Reviewer_gNAR · 2026-04-03
> >
> > My concerns have been addressed. The additional experiments and runtime analysis make the empirical case significantly stronger.

---

> > > ### Author Response · Authors · 2026-04-03
> > >
> > > We sincerely thank the reviewer for the constructive feedback and for confirming that the concerns have been fully resolved. The suggestions around profiling directly improved the paper, and we are grateful for the time and care invested in the review.

---

### Official Review · Reviewer_aaZv · 2026-03-08

**Soundness:** 4
**Presentation:** 3
**Significance:** 3
**Originality:** 3
**Overall Recommendation:** 4
**Confidence:** 4

**Summary:**

This paper connects contrastive learning losses with the negative log likelihood of the top1 choice. Different extreme-value geometries correspond to different ways of transforming the logits. This paper highlights that the bounded nature of the logits (cosine similarities) corresponds to the Weibull tail rather than the Gumbel tail as InfoNCE assumes. To fix this gap, it proposes WEINCE, which interpolates between Plackett-Luce logits and a Weibull shortfall logit, with the mixing weights estimated from the batch statistics.

**Compliance With Llm Reviewing Policy:**

Affirmed.

**Final Justification:**

My main concerns are addressed in the rebuttal, and I think the other suggestions can be addressed in the authors' next steps (adding dedicated sensitivity analysis and other applications) sound good. Therefore, I will retain my original positive score.

**Key Questions For Authors:**

Please refer to the "Strengths And Weaknesses" section where I listed my questions.

**Limitations:**

Yes, this paper has discussed the potential future work directions.

**Strengths And Weaknesses:**

**Strength:**
- Idea is insightful and presented well: It is insightful to connect softmax to a discrete choice model with Gumbel noise. Figure 1, Algorithm 1, and Table 1 are clear and enjoyable to read.
- Method is clean and easy to use: This paper provides a heuristic to estimate $\lambda_i$ and $\beta$ rather than setting them as hyperparameters.

**Questions/Suggestions/Weakness:**

1. Can the authors please specify how the hyperparameters $K_{\text{tail}}, \rho_0, m, \kappa_{\rho}, \kappa_{\text{AIC}}, \varepsilon$ are picked in the experiments, and what is the search range of each of them? Are the experimental results sensitive to these hyperparameters’ values? It is also nice to add a section providing intuition and actionable guidelines on choosing their values.
2. Currently, the performance is limited to ResNet, and the most challenging task is CIFAR-100. It would be nice to additionally report performance on larger datasets (such as Tiny-ImageNet) and vision transformer models. I observe the message that WEINCE's improvement is more pronounced on challenging datasets, so I am curious about Tiny-ImageNet results.
3. It would also be nice to add experiments on fine-tuning (a small version of) CLIP, which is a relevant and important modern application of contrastive loss.

---

> ### Author Rebuttal · Authors · 2026-03-31
>
> We thank the reviewer for the favorable assessment and for the specific, constructive suggestions. We are glad the reviewer found the connection between softmax and discrete choice models insightful and the method clean and easy to use. We especially appreciate the observation that the improvement is more pronounced on the more challenging datasets, as we had not really paid attention to that, and the new results we obtained seem to point in that direction too.
>
> ---
>
> **Q1: Hyperparameter selection and sensitivity.**
> *Can the authors please specify how the hyperparameters are picked, and what is the search range of each of them?*
>
> Algorithm 1 introduces six selector hyperparameters.
>
> - $K_{\text{tail}}$: controls the "locality" of the tail fit (how many of the smallest shortfalls are used). We set this as ~10% of negatives with a floor of 32 points, and do not swipe it.
> - $\rho_0$: the cap-proximity pivot, implemented as the 20th-percentile of per-anchor $\rho_i$ values. Controls how many anchors count as "near cap."
> - $m$: AIC margin. Larger $m$ requires stronger Weibull evidence before activating. Default: $m{=}2$.
> - $\kappa_\rho$, $\kappa_{\text{AIC}}$: gate sharpness for the cap-proximity and AIC signals respectively. Defaults: $\kappa_\rho{=}10$, $\kappa_{\text{AIC}}{=}1$.
> - $\varepsilon$: numerical clipping constant ($10^{-6}$), chosen heuristically for stability.
>
> We explored small validation grids $m \in \{1, 2, 4\}$, $\kappa_\rho \in \{5, 10, 20\}$, $\kappa_{\text{AIC}} \in \{0.5, 1, 2\}$. All configurations were selected using the validation split only, never the test set.
>
> Regarding sensitivity, the selector was most sensitive to the activation frequency of the correction, controlled by $\rho_0$ (via the quantile) and $m$. The sharpness parameters $\kappa_\rho$, $\kappa_{\text{AIC}}$ are second-order controls. $\varepsilon$ and $K_{\text{tail}}$ had negligible effect in our runs.
>
> We acknowledge that a formal ablation table would strengthen this discussion. We prioritized other rebuttal experiments (new datasets, architectures, and modalities) but will include a dedicated sensitivity analysis for the camera-ready version. We will also include a discussion and suggestions on how to select the parameters.
>
> ---
>
> **Q2: Larger datasets and vision transformers.**
> *Currently, the performance is limited to ResNet, and the most challenging task is CIFAR-100. It would be nice to additionally report performance on larger datasets (such as Tiny-ImageNet) and vision transformer models.*
>
> We would first like to note that the submission also reports results on ImageNet-32 (1,000 classes, the full ImageNet at 32×32, +0.97% linear accuracy with ResNet-18), which is more complex than CIFAR-100.
>
> As the reviewer anticipated, WEINCE's improvement is indeed more pronounced on harder tasks. During the rebuttal we added Tiny-ImageNet (200 classes, 64×64) with both ResNet-18 and ViT-Small:
>
> | Backbone | Method | Lin. Acc | kNN Acc | R@1 | R@2 | R@5 | R@10 | R@20 |
> |:---|:---|:---:|:---:|:---:|:---:|:---:|:---:|:---:|
> | ResNet-18 | InfoNCE | **30.76** | 24.41 | 17.29 | 25.29 | 38.26 | 49.87 | 61.82 |
> | ResNet-18 | WEINCE | 30.66 | **25.13** | **18.84** | **26.64** | **39.12** | **50.14** | **61.90** |
> | ViT-Small | InfoNCE | 33.12 | 25.92 | 19.30 | 27.29 | 40.36 | 51.17 | 62.63 |
> | ViT-Small | WEINCE | **36.53** | **31.81** | **23.53** | **32.04** | **45.15** | **55.96** | **66.97** |
>
> The ViT-Small kNN gain (+5.89%) and R@1 gain (+4.23%) are the largest across all our benchmarks, consistent with the reviewer's intuition that the endpoint effect should amplify on more challenging tasks. Gains concentrate at small $k$ (R@1, R@2), which is the regime where near-ceiling negatives matter most.
>
> ---
>
> **Q3: Fine-tuning CLIP.**
> *It would also be nice to add experiments on fine-tuning (a small version of) CLIP.*
>
> Thank you for this suggestion. Although we did not have time during the rebuttal period to run CLIP experiments, we have confirmed that the shortfall correction generalizes beyond SimCLR to other frameworks and modalities with language. Specifically, we tested on unsupervised SimCSE (BERT-base, +3.53 Spearman on STS-B) and on RLHF reward modeling with a Plackett–Luce ranking loss (Nectar dataset, DeBERTa-v3 RM, +4.5 pp top-1 accuracy). Full details and tables are in our response to Reviewer gNAR. CLIP fine-tuning is a natural next step that we plan to pursue, and we expect the same EVT argument to apply since CLIP uses the same InfoNCE + cosine similarity structure.

---

> > ### Author Rebuttal · Reviewer_aaZv · 2026-04-02
> >
> > Thank the authors for the rebuttal. My concerns are addressed, and I think the next steps (dedicated sensitivity analysis and other applications) sound good. Therefore, I will retain my positive score.

---

> > > ### Author Response · Authors · 2026-04-02
> > >
> > > We sincerely thank the reviewer for the thoughtful and constructive feedback throughout this process. Your suggestion to test on Tiny-ImageNet proved valuable.

---

### Official Review · Reviewer_4Nq3 · 2026-03-12

**Soundness:** 3
**Presentation:** 3
**Significance:** 3
**Originality:** 3
**Overall Recommendation:** 4
**Confidence:** 4

**Summary:**

This paper proposes an improved contrastive learning loss function called WEINCE (Weibull Enhanced InfoNCE), designed to address the extreme sample problem in the traditional InfoNCE loss when dealing with bounded similarities (such as cosine similarity). By introducing extreme value theory, specifically the Weibull distribution, WEINCE can effectively handle situations where negative sample similarities approach their maximum values, thereby enhancing the performance of contrastive learning. Experiments validate the effectiveness of this method on datasets such as CIFAR-10 and STL-10. Particularly on network architectures like ResNet18 and ResNet50, WEINCE demonstrates superior performance compared to traditional InfoNCE in downstream linear classification and kNN evaluation tasks.

**Compliance With Llm Reviewing Policy:**

Affirmed.

**Final Justification:**

The authors have almost addressed my core concerns. I believe the score can be revised to "weak accept."

**Key Questions For Authors:**

1.The experimental validation in the paper is primarily conducted on relatively small-scale datasets such as CIFAR-10 and STL-10. This makes it difficult to ascertain whether the proposed method can generalize to more challenging tasks and real-world scenarios. Have the authors considered evaluating the proposed method on larger-scale or more complex datasets?

2.The WEINCE loss function improves upon the InfoNCE loss by introducing extreme value theory, which appears to be an incremental optimization of existing methods. Could the authors further elaborate on why this improvement should be regarded as a breakthrough innovation rather than a technical adjustment? Are there any existing contrastive learning methods that share a similar core idea? Compared to prior methods, what specific and unique advantages does WEINCE offer? What specific issues that previous methods failed to address does it actually solve?

3.The derivation involving extreme value theory in the paper is mathematically quite complex. Could the authors provide a more intuitive explanation or visual comparison to bridge the gap between this theory and the behavior of existing loss functions? Specifically, for readers without a strong mathematical background, how can they better understand the core intuition behind WEINCE, and why extreme value theory is particularly suitable for handling bounded similarity problems?

4.The current experiments mainly compare WEINCE with the standard InfoNCE loss. Could the authors consider including comparisons with a broader range of contrastive learning methods to better contextualize the empirical advantages of the proposed approach?

**Limitations:**

“No”

1.The efficiency and scalability issues of the WEINCE loss function when handling large-scale datasets.

2.The applicability limitations of the method across different domains, particularly its performance in non-contrastive learning tasks.

**Strengths And Weaknesses:**

Strengths:

Soundness:
The paper's core theoretical contribution lies in introducing extreme value theory into the design of contrastive learning loss functions, providing a theoretically grounded solution for handling bounded similarity. The theoretical derivation is generally rigorous.

Presentation:
The paper is clearly written and well-structured, effectively conveying the research motivation and core ideas to the reader.

Significance:
The problem addressed—"handling hard negative samples"—holds practical relevance in the field of contrastive learning. Solving this issue has potential value for improving model robustness.

Originality:
Combining extreme value theory with the InfoNCE loss represents a technically distinctive approach within contrastive learning. Its novelty lies in introducing a new theoretical perspective to improve upon an existing method.

Weaknesses:

Soundness:
The sufficiency of experimental validation is a major shortcoming of the paper. The experimental setup is quite limited, evaluating only on small-scale architectures like ResNet18/50 and simple, small-scale datasets such as CIFAR-10/STL-10. Consequently, the effectiveness and generalizability of the proposed WEINCE loss on more complex, large-scale tasks remain inadequately demonstrated, weakening the empirical support for its theoretical claims.

Presentation:
Despite the overall clarity, the paper relies heavily on complex mathematical formulations when elaborating its core theory (extreme value theory), lacking intuitive explanations and illustrative examples. This increases the reader's comprehension barrier, making it difficult, especially for those unfamiliar with the theory, to quickly grasp its core mechanism and its intuitive connection to the problem being solved.

Significance:
The paper's potential impact is constrained by its insufficient experimental validation. As its effectiveness has not been thoroughly verified on widely recognized benchmarks or tasks, it is currently difficult to assess its true value for future research or practical applications. Therefore, its contribution currently remains at the level of technical refinement, with limited influence.

Originality:
The paper's contribution is incremental; it essentially proposes a fix for a specific limitation (handling bounded similarity) of the existing InfoNCE loss. While incorporating extreme value theory is a novel angle, the overall novelty resides in a nuanced adjustment to existing techniques, rather than proposing a new learning paradigm or achieving a conceptual breakthrough.

---

> ### Author Rebuttal · Authors · 2026-03-31
>
> We thank the reviewer for recognizing the soundness of the theoretical derivation and the novelty of the EVT perspective. Different from other reviews, we have allocated most of our space here for Q3 and created more visuals and intutive explanations.
>
> ---
>
> **Q1: Generalization to larger-scale datasets.**
>
> Please note that the submission already includes CIFAR-100 (100 classes, +4.82% lin acc R50) and ImageNet-32 (1,000 classes, +0.97%). During the rebuttal we extended the evaluation substantially. We ran our experiments on a new dataset (Tiny-ImageNet), on a different architecture (ViT), and on a new modality (SimCSE). Across 9 configurations spanning 3 domains (vision, NLP, RLHF) and 3 architecture families (ResNet, ViT, BERT/DeBERTa), WEINCE improves or matches the baseline in all but one case (RoBERTa). The ViT-Small result confirms the correction is not architecture-specific. The SimCSE and RLHF results confirm cross-domain generality. For full tables and details, see our response to Reviewer gNAR.
>
> ---
>
> **Q2: Why this is more than an optimization.**
>
> - We identify a structural misspecification in the most widely used contrastive objective. Also see the response to next question and figures we have created there.
> - The practical fix and theoretical insight potentially apply to any softmax-based ranking loss over bounded scores, as we now demonstrate across vision, NLP, and RLHF. This can have many applicatons across domains.
> - We bring a minimal and well-justified solution to the question of how to treat harder negatives differently, *without explicitly treating them differently.*
>
> ---
>
> **Q3: Intuitive and visual explanation of the EVT argument.**
>
> We have included 4 figures throughout our response here.
>
> Key idea: Bounded similarities need a bounded-tail model, and softmax is not it.
>
> InfoNCE can be interpreted a statistical model of who wins a competition. Each candidate $j$ has score $U_j = s_j + \varepsilon_j$ with $\varepsilon_j \sim \text{Gumbel}(0,\tau)$, and the winner is $\arg\max_j U_j$. Under Gumbel noise the winning probability is the softmax $p_k \propto \exp(s_k/\tau)$. The Gumbel has infinite support, so this model treats the score space as if it extends indefinitely beyond the cosine ceiling of 1.
>
> The consequence is that model is "not surprised enough" by hard negatives near the cap. Since the Gumbel tail assigns density beyond $s=1$ (see the [POT density plot](https://anonymous.4open.science/r/rebuttal_figures_repo-DFEF/images/pot_fit.png), red curve vs green cap line), a negative with $s=0.95$ is "just another score" to the Gumbel model. Under the correct Weibull model, which respects the finite ceiling, $s=0.95$ is extreme because almost no probability mass remains above it. The [QQ plot](https://anonymous.4open.science/r/rebuttal_figures_repo-DFEF/images/pot_fit_qq.png) confirms this. Fitting the upper tail of scores from a frozen encoder yields $\hat{\xi}=-0.39$ (Weibull), and the Gumbel model systematically overestimates the upper quantiles.
>
> EVT is used to make this precise. The FTG theorem says there are exactly three possible tail types for maxima: Gumbel (unbounded), Fréchet (heavy-tailed), Weibull (finite endpoint). For bounded distributions, the extremes must be Weibull. Under Weibull tails the winning probability becomes $p_k \propto (1-s_k)^{-\beta}$, which concentrates much more sharply on near-ceiling negatives than $\exp(s_k/\tau)$.
>
> Because the Gumbel model under-reacts to hard negatives, it allocates gradient to the wrong places. We verify this directly by freezing an encoder, repeatedly sample $K$-subsets, and compare where failures come from (risk, black) with where each loss puts its gradient (softmax red, Weibull blue). The InfoNCE gradient w.r.t. negative $j$ is $p_j^{SM}/\tau$, where $1/\tau$ is constant across negatives. So the normalized softmax gradient is the model's predicted win distribution. If the Gumbel assumption is correct, this must match the empirical risk and divergence from it is an evidence of misspecification.
>
> The [score-binned figure](https://anonymous.4open.science/r/rebuttal_figures_repo-DFEF/images/risk_grad_score.png) shows softmax gradient spreads mass across easy negatives (where risk is zero), while the Weibull gradient tracks the risk. The [alignment figure](https://anonymous.4open.science/r/rebuttal_figures_repo-DFEF/images/risk_grad_alignment.png) shows that Weibull achieves >0.85 cosine alignment with risk (vs <0.2 for softmax), and the gap widens with increasing $K$, as our theory predicts.
>
> **Q4: Comparisons with other contrastive methods.**
>
> We think contribution is orthogonal to methods that change negative sampling, or architecture (MoCo, BYOL). WEINCE modifies only the logit computation inside the cross-entropy loss, so it can technically be combined with any of these. We decided to spend the rebuttal time addressing the other questions and didn't have the resources to run the experiments on other architectures.

---

> > ### Author Rebuttal · Reviewer_4Nq3 · 2026-04-05
> >
> > The authors have almost addressed my core concerns. I believe the score can be revised to "weak accept."

---

> > > ### Author Response · Authors · 2026-04-05
> > >
> > > Dear Reviewer 4Nq3,
> > >
> > > Thank you very much for engaging with our rebuttal and for your thoughtful acknowledgement.
> > >
> > > We're glad the concerns have been addressed. We just wanted to flag that, for the score revision to be reflected officially, **the numerical score in the original review itself needs to be edited**. The acknowledgement comment alone may not be picked up by the system. We wanted to mention this in case it's helpful, as the discussion period closes soon. Thank you again for your time and constructive feedback.
> > >
> > > Best Regards,
> > > Authors

---

### Official Review · Reviewer_8pFv · 2026-03-12

**Soundness:** 4
**Presentation:** 3
**Significance:** 3
**Originality:** 3
**Overall Recommendation:** 5
**Confidence:** 3

**Summary:**

This work proposes a drop-in replacement loss function for the InfoNCE loss classically used in contrastive learning. The authors base their study on the observation of the following mismatch: the InfoNCE loss is based on a top-1 probability model where the underlying noise distribution is a Gumbel, while classical contrastive learning set-ups use a cosine similarity metric, which imposes a maximum value on the similarity values.

To circumvent this mismatch, the authors use Extreme Value Theory (EVT) results, and in particular the Fisher-Tippett-Gnedenko Theorem, which characterizes a canonical model of bounded distributions as Weibull distributions. They use this insight to modify the InfoNCE loss by changing the cosine similarity by a convex combination with the negative logarithm of the distance to the endpoint ($1$ in the case of cosine endpoint). They experimentally verify that the estimated convex parameter is non-zero, which shows the relevance of their model.

Finally, they propose the WEINCE loss as follows: for a given cosine similarity $s_{ij}$, use the softmax loss, not on the temperature scaled $s_{ij}/\tau$ (as INCE would) but on a convex combination with parameter $\lambda\in [0,1]$ between $s_{ij}/\tau$ and the negative logarithm of the distance to the right-end point ($1$ in the case of cosine similarity), ie $-\beta\log(1-s_{ij})$. In particular, their method requires two parameters $\lambda, \beta$, which they explain how to measure online using the data. They finally experimentally validate their method in different settings, including image datasets and show superior performance compared with the baseline.

**Compliance With Llm Reviewing Policy:**

Affirmed.

**Final Justification:**

This work is overall well-presented with sound results and both interesting theory and empirical results supporting the theory. The empirical results are strong and the perspective of EVT to derive a new loss function is very original and interesting.

The authors answered my main questions about behavior at scale of their method with convincing additional experiments. They additionally stated that they would improve presentation of the manuscript with "informal statements of the key theorems and these figures" as was proposed to improve clarity and intuition.

I recommend acceptance.

**Key Questions For Authors:**

1) Why did you choose to report the k-Nearest Neighbour recall and downstream linear evaluation accuracy as metrics in your experiments? Did you think of reporting other evaluation metrics, or why are these sufficient?

2) How do you expect your proposed method to behave at larger scales? It seems to me that the concentration of similarities to the endpoint should amplify at scale, hence rendering your method more efficient.

3) Did you think of other methods than a convex combination in 6.1 to incorporate a Weibull model in the loss?

**Limitations:**

The authors don't discuss limitations nor provide an impact statement.

A potential limitation of this method would be some experimental settings where the similarities, in fact, do not concentrate on the endpoint, rendering the proposed method less relevant. I was wondering if the authors had thought about such cases?

**Strengths And Weaknesses:**

**Strengths**

*Soundness* This work is very well structured, and the results are derived in a sound and rigorous way. The authors first identify a mismatch between the loss usually used in contrastive learning and the boundedness of the cosine similarities. They formalize the problem using EVT and an important Theoretical result of the field in Theorem 4.1. They then propose a modification of the loss based on this result and experimentally verify the relevance of their modification by showing that the new hyperparameter $\lambda$, which they introduced, is optimally non-zero. Finally, they propose a new algorithm, which is rigorously stated in Algorithm 1 and show improved results in numerous experiments, Tables 2 and 3. Finally, the theoretical results are reported in Appendices A and B, although I did not fully check all of them.

*Originality* The use of EVT to derive a formulation for a new contrastive loss with a Weibull component is novel and interesting. It combines theoretical tools with a practical implementation in an elegant way to address an important mismatch of the current loss used in practice.

*Significance* The proposed method is outperforming the baseline by a strong margin, with empirical results superior to the baseline in all experiments (see Table 2,3). Additionally, while the proposed method combines in one way (convex combination) Weibull and Placket-Luce noise models, some different ways of incorporating a Weibull component into the loss may be interesting and could be pursued in future works. The authors additionally highlight numerous interesting future directions in their conclusion.

**Weaknesses**

*Presentation* Overall, the results are easy to follow and well-explained. However, a few important details are postponed to the appendix, which made the main paper not completely self-contained. Incorporating at least an intuition of these without going too much into the details could help make the story more fluid. For example, Theorem 4.3 refers to a theorem (A.2) and a set of assumptions (B.1) stated in the appendix, without an informal intuition given prior. On the other hand, I appreciated the author's effort to give an intuition on how $\lambda_i,\beta_i$ are chosen in section 6, although the exact formulations are explained in the appendix. Finally, a minor issue is that Figure 6 is a bit small and hard to read.

---

> ### Author Rebuttal · Authors · 2026-03-31
>
> We thank the reviewer for the careful reading and strong endorsement of the work. We are glad the reviewer found the use of EVT to derive the Weibull correction both novel and elegant. Aside from what has been asked, we have also run additional experiments. We gave more details in our response to Q2, and even further details in our response to Reviewer gNAR.
>
> ---
>
> **Q1: Choice of evaluation metrics (kNN recall and linear accuracy).**
>
> These are the two standard evaluation protocols for self-supervised contrastive encoders. Linear probing measures how linearly separable the frozen features are and has been widely adopted following e.g. [Wang & Isola, 2020](https://arxiv.org/abs/2005.10242). kNN recall measures local neighborhood quality without any learned classifier, as used in e.g. [Feng & Patras, 2023](https://arxiv.org/abs/2303.12756). Together they assess complementary aspects of representation quality.
>
> ---
>
> **Q2: Expected behavior at larger scale.**
>
> We agree with the reviewer's intuition and have now confirmed it empirically with two new ablations on frozen encoders.
>
> In the first experiment ([figure](https://anonymous.4open.science/r/rebuttal_figures_repo-DFEF/images/lambda_diff_fixedKneg.png)), we fix the number of hard negatives in the choice set to $K_{\text{neg}}{=}64$ and vary the size of the negative pool $K$ from which they are selected. As $K$ grows, the top $K_{\text{neg}}$ negatives become more extreme (smaller shortfalls), and we observe a strong monotone increase in both the fitted $\hat{\lambda}$ and the held-out log-likelihood gain $\Delta$, with $\hat{\lambda}$ approaching 1 for large $K$. This matches the EVT prediction since bounded endpoint effects become more pronounced as the available pool grows.
>
> In the second experiment ([figure](https://anonymous.4open.science/r/rebuttal_figures_repo-DFEF/images/lambda_diff_fixedK.png)), we do the reverse by fixing the pool at $K{=}512$ and varying $K_{\text{neg}}$. As $K_{\text{neg}}$ increases, the choice set includes progressively less extreme negatives, so both $\hat{\lambda}$ and $\Delta$ decrease. This confirms that the Weibull correction is primarily needed for the most competitive near-cap negatives.
>
> Together these results support the reviewer's prediction that WEINCE should become more effective at larger scale.
>
> We would also like to note that we have run additional experiments. We ran our experiments on a new dataset (Tiny-ImageNet), on a different architecture (ViT), and on a new modality (SimCSE). Across 9 configurations spanning 3 domains (vision, NLP, RLHF) and 3 architecture families (ResNet, ViT, BERT/DeBERTa), WEINCE improves or matches the baseline in all but one case (RoBERTa). The ViT-Small result confirms the correction is not architecture-specific. The SimCSE and RLHF results confirm cross-domain generality. In particular, by inlcuding more challenging datasets, and observing that (pointed out by Reviewer aaZv) our method provides better improvements on more challenging datasets. For full tables and details, see our response to Reviewer gNAR.
>
> ---
>
> **On presentation.** We appreciate the suggestion to add more intuition for the theoretical results in the main text. We have prepared new diagnostic figures (POT tail analysis and a gradient misallocation visualization, described in our response to Reviewer 4Nq3) that make the EVT argument visual. We will incorporate informal statements of the key theorems and these figures into the revised paper, and we will enlarge Figure 6.
>
> **On limitations.** The reviewer correctly identifies the key limitation that when similarities do not concentrate near the endpoint, the correction becomes irrelevant. This is both by theory, and also in practice, since $\hat{\lambda} \to 0$. This is by design, as the selector parameter deactivates gracefully. We will add an explicit limitations discussion and impact statement in the revision.

---

> > ### Author Rebuttal · Reviewer_8pFv · 2026-04-03
> >
> > I thank the authors for their careful reply to my questions and concerns. These have especially been resolved as follows:
> >
> > **Expected behavior at a larger scale.** I appreciate the authors' additional experiments confirming that WEINCE becomes more effective at larger scales. I appreciate that it follows intuition, strengthening the soundness of the results.
> >
> > **Limitations** The reviewers acknowledged the outlined limitation and committed to "add an explicit limitations discussion and impact statement in the revision".
> >
> > **Presentation** I appreciate that the authors will "incorporate informal statements of the key theorems and these figures into the revised paper".
> >
> > Finally, the additional experiments conducted by the authors strengthen the soundness and significance of their method. I choose to keep my score of 5 and recommend acceptance.

---

> > > ### Author Response · Authors · 2026-04-03
> > >
> > > Thank you for your thoughtful engagement with our work throughout the review process. We are grateful for your continued support and recommendation for acceptance.

---

### Decision · Program_Chairs · 2026-04-30

**Decision:**

Accept (regular)

**Comment:**

This paper revisits the InfoNCE objective from a statistical perspective, interpreting it as a discrete choice model with Gumbel noise, and argues that this implicit assumption can be mismatched in settings with bounded similarities. Using insights from extreme value theory, the authors motivate a modification of the loss that accounts for Weibull-type tail behavior, and propose the resulting WEINCE objective.

The reviewers agree that the paper is well-written, technically sound, and introduces an interesting and well-motivated perspective on contrastive learning. The combination of a clear statistical interpretation with a simple, practical modification of the loss is a strength, and the empirical results consistently show improvements over standard InfoNCE across several settings. The rebuttal further strengthened the paper by providing additional experiments and clarifying the scope and limitations of the approach.

At the same time, the contribution is best understood as a targeted refinement of existing contrastive objectives rather than a fundamentally new framework. The proposed modification is most relevant in regimes where similarities concentrate near their upper bound, and while the empirical results are encouraging, they remain somewhat limited in scale. These aspects were appropriately discussed during the review process.

Overall, the paper provides a neat and practically relevant contribution, combining a well-justified modeling insight with a simple and effective method, and I recommend it for acceptance.